# Tissue environment, not ontogeny, defines murine intestinal intraepithelial T lymphocytes

**Alejandro J Brenes[1,2†], Maud Vandereyken[3†], Olivia J James[3], Harriet Watt[3], Jens Hukelmann[1], Laura Spinelli[2], Dina Dikovskaya[3], Angus I Lamond[1], Mahima Swamy[2,3]***

[1]Centre for Gene Regulation and Expression, University of Dundee, Dundee, United Kingdom; [2]Division of Cell Signalling and Immunology, School of Life Sciences, University of Dundee, Dundee, United Kingdom; [3]MRC Protein Phosphorylation and Ubiquitylation Unit, University of Dundee, Dundee, United Kingdom

**Abstract** Tissue-resident intestinal intraepithelial T lymphocytes (T-IEL) patrol the gut and have important roles in regulating intestinal homeostasis. T-IEL include both induced T-IEL, derived from systemic antigen-experienced lymphocytes, and natural T-IEL, which are developmentally targeted to the intestine. While the processes driving T-IEL development have been elucidated, the precise roles of the different subsets and the processes driving activation and regulation of these cells remain unclear. To gain functional insights into these enigmatic cells, we used high-resolution, quantitative mass spectrometry to compare the proteomes of induced T-IEL and natural T-IEL subsets, with naive CD8$^+$ T cells from lymph nodes. This data exposes the dominant effect of the gut environment over ontogeny on T-IEL phenotypes. Analyses of protein copy numbers of >7000 proteins in T-IEL reveal skewing of the cell surface repertoire towards epithelial interactions and checkpoint receptors; strong suppression of the metabolic machinery indicating a high energy barrier to functional activation; upregulated cholesterol and lipid metabolic pathways, leading to high cholesterol levels in T-IEL; suppression of T cell antigen receptor signalling and expression of the transcription factor TOX, reminiscent of chronically activated T cells. These novel findings illustrate how T-IEL integrate multiple tissue-specific signals to maintain their homeostasis and potentially function.

**\*For correspondence:**
m.swamy@dundee.ac.uk

†These authors contributed equally to this work

**Competing interest:** The authors declare that no competing interests exist.

## Introduction

The presence of tissue-resident immune cells enables a quick response to either local stress, injury, or infection. Understanding the functional identity of immune cells and their shaping by the tissue environment is therefore critical to understanding tissue immunity. Tissue-resident intestinal intraepithelial T lymphocytes (T-IEL) reside within the intestinal epithelium and consist of a heterogenous mix of natural and induced T-IEL (*Olivares-Villagómez and Van Kaer, 2018*). All T-IEL express a T cell antigen receptor (TCR), consisting of either αβ or γδ chains, alongside TCR co-receptors, that is, CD8αβ or CD8αα and to a lesser extent CD4(+/-). The most prevalent IEL subsets within the epithelium of the murine small intestine are derived directly from thymus progenitors, the so-called natural or unconventional T-IEL. These natural T-IEL express either TCRγδ and CD8αα (TCRγδ CD8αα T-IEL), which account for ~50 % of the total T-IEL pool, or express TCRαβ and CD8αα (TCRαβ CD8αα T-IEL), which account for ~25 % of the total T-IEL. TCRβ CD8αα T-IEL are derived from CD4⁻CD8⁻ double negative (DN) progenitors in the thymus by agonist selection. Conversely, induced T-IEL are antigen-experienced, conventional CD4$^+$ or CD8αβ$^+$ αβ T cells that are induced to establish tissue residency within the intestinal epithelium, most likely in response to cues from dietary antigens and the

microbiota, as evidenced by a strong reduction in their numbers in germ-free and protein antigen-free mice (*Di Marco Barros et al., 2016*). These induced T-IEL (TCRβ CD8αβ T-IEL) are believed to have substantial overlap with tissue-resident memory T (T_{RM}) cells (*Sasson et al., 2020*) and are present in high numbers in human intestines. How these induced T-IEL are formed, their functional importance, and the role of the gut environment in deciding their fate are still the focus of intense study.

Residing at the forefront of the intestinal lumen, T-IEL are exposed to a range of commensal bacteria and their metabolites, dietary metabolites and antigens, and potential pathogens. These immune cells are therefore faced with the conflicting tasks of protecting the intestinal barrier, while also preventing indiscriminate tissue damage. Previous gene expression studies have identified T-IEL as having an 'activated-yet-resting' phenotype, with the expression of several activation markers, such as Granzymes and CD44, along with inhibitory receptors, such as the Ly49 family and CD8αα (*Denning et al., 2007*; *Fahrer et al., 2001*; *Shires et al., 2001*). Yet, it is still unclear how T-IEL are kept in check at steady state (*Vandereyken et al., 2020*). T-IEL effector responses can get dysregulated in chronic inflammatory conditions, such as celiac disease and inflammatory bowel diseases, therefore we need insight into the regulation of these cells. Moreover, we lack an understanding of how T-IEL are programmed to respond to specific epithelial signals, and how this is dictated and regulated by the tissue microenvironment.

In this study, we use quantitative proteomics to explore the differences between induced T-IEL and systemic T cells from lymph nodes (LN), from which induced T-IEL are ostensibly derived. We also compare induced T-IEL with the natural TCRγδ and TCRαβ T-IEL subsets in the gut. Our findings suggest that the tissue environment largely overrides any developmental imprinting of the cells to define the proteomic landscape of intestinal-resident T-IEL, and reveal important metabolic and protein translation constraints to T-IEL activation. Importantly, we also uncover evidence of chronic T cell activation potentially driving a partially exhausted phenotype in both the induced and natural T-IEL subsets.

## Results

### Tissue microenvironment defines T-IEL as distinct from systemic T cells

CD8[+] T-IEL subsets were purified from wild-type (WT) murine small intestinal epithelial preparations to >95 % purity by cell sorting (*Figure 1—figure supplement 1*). Next, high-resolution mass spectrometry (MS) was performed to obtain an in-depth characterisation of the proteomes of the three main CD8[+] T-IEL subsets in the intestine. Tandem mass tags (TMT) were used with synchronous precursor selection (SPS) to obtain the most accurate quantifications for all populations (*Figure 1a*). To evaluate how T-IEL related to other immune populations, we first compared the proteomes of T-IEL with other TMT-based proteomes of various T cell populations currently available within the Immunological Proteome Resource (ImmPRes, http://immpres.co.uk), an immune cell proteome database developed in-house (*Howden et al., 2019*). Even though T-IEL are thought to have an effector-like phenotype, by using principal component analysis (PCA), we found that T-IEL were much more similar to ex vivo naïve CD8[+] T cells, than to in vitro activated, effector cytotoxic T cells (CTL) (*Figure 1b*). Hence, we did an in-depth, protein-level comparison of the three T-IEL subsets with two naïve CD8[+] T cells from the LN. The two LN naïve CD8[+] T cells used here were derived either from WT mice, similar to the T-IEL, or from P14 transgenic mice, which express a TCR specific for a peptide derived from lymphocytic choriomeningitis virus. P14 T cells were included as a genetically and developmentally distinct comparator for WT LN T cells. To enable cross-comparisons, all five populations were acquired using the same TMT-based SPS-MS3 method and they were all analysed together using MaxQuant (*Cox and Mann, 2008*). The data were searched using a 1 % false discovery rate (FDR) at the protein and peptide spectrum match (PSM) level (for more details, see Materials and methods), both raw and processed data were uploaded to PRIDE (*Perez-Riverol et al., 2019*) under accession PXD023140. The dataset provided an in-depth overview of the proteome, with over 8200 proteins detected in total, where each of the five populations showed similar coverage, ranging from 6500 to 7500 proteins detected in all of them (*Figure 1c*).

For all downstream analyses we converted the raw MS intensity values into estimated protein copy numbers using the 'proteomic ruler' (*Wisniewski et al., 2014*). First, these copy numbers were used to estimate the total protein content for all five populations, which revealed no major differences in

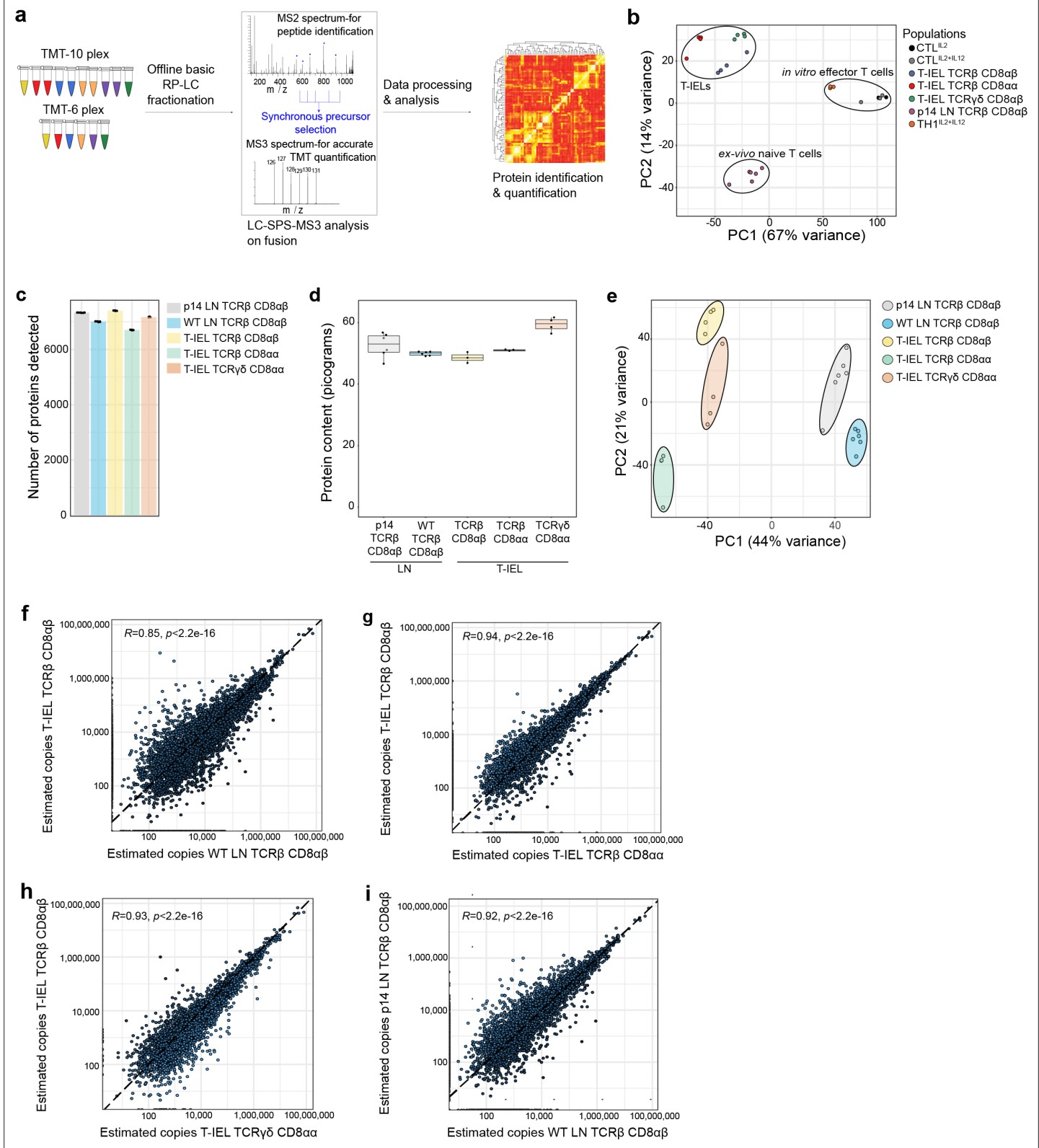

**Figure 1.** Quantitative proteomic analyses of induced and natural tissue-resident intestinal intraepithelial T lymphocytes (T-IEL) subsets. (**a**) Schematic of the mass spectrometry (MS)-based proteomics workflow. The data were acquired at the MS3 level with synchronous precursor selection (see Materials and methods). (**b**) Principal component analysis comparing the tandem mass tags (TMT)-based estimated protein copy numbers of conventional naïve and effector T cells with T-IEL. CTL, cytotoxic T lymphocytes. (**c**) Bar plot showing the number of proteins identified across all replicates in the five

*Figure 1 continued on next page*

*Figure 1 continued*

populations used for this study. (**d**) Box plot showing the MS-based protein content estimation for all replicates used across the five populations. (**e**) Principal component analysis comparing the estimated protein copy numbers across conventional naïve lymph node (LN) T cells and T-IEL subsets. (**f–i**) Scatter plot comparing the estimated copy numbers for (**f**) TCRαβ CD8αβ T-IEL and wild-type (WT) LN TCRαβ CD8αβ T cells, (**g**) TCRαβ CD8αβ T-IEL and TCRαβ CD8αα T-IEL, (**h**) TCRαβ CD8αβ T-IEL and TCRγδ CD8αα T-IEL, (**i**) WT LN TCRαβ CD8αβ T cells and P14 LN TCRαβ CD8αβ T cells. Pearson correlation coefficient are included within all the scatter plots. The proteomics data displayed on the plots include CTL (n = 3 biological replicates), conventional naïve LN T cells (both WT and P14 n = 6 biological replicates), TCRαβ CD8αβ T-IEL and TCRαβ CD8αα T-IEL (n = 3 biological replicates) and TCRγδ CD8αα T-IEL (n = 4 biological replicates). For box plots, the bottom and top hinges represent the first and third quartiles. The top whisker extends from the hinge to the largest value no further than 1.5× IQR from the hinge; the bottom whisker extends from the hinge to the smallest value at most 1.5× IQR of the hinge. The bar plots show the mean. Total number of proteins identified and total protein content across all populations are available in *Figure 1—source data 1*.

The online version of this article includes the following figure supplement(s) for figure 1:

**Source data 1.** Total protein identifications and total protein content across all populations.

**Figure supplement 1.** Gating strategy used to identify and isolate tissue-resident intestinal intraepithelial T lymphocytes (T-IEL) subsets using fluorescence activated cell sorting (FACS).

most, except for the TCRγδ CD8αα T-IEL, which displayed a slightly higher protein content than the rest (*Figure 1d*). Next, the copy numbers were used as input for a second dimensionality reduction analysis via PCA, focussed now on comparing the T-IEL and LN populations. The results indicated that across the first component, which explains 44 % of variance, there was a clear separation between T-IEL and LN populations (*Figure 1e*), highlighting that the three T-IEL subsets share much closer identity to each other, than to the naïve LN T cell populations.

To explore these results further we compared each population to each other. As induced TCRαβ CD8αβ T-IEL are thought to be derived from systemic T cells that respond to antigen in organised lymphoid structures, and then migrate into intestinal tissues, we first compared their proteome to the systemic WT LN TCRαβ CD8αβ T cells. Unexpectedly, the Pearson correlation coefficient comparing the estimated protein copy number of TCRαβ CD8αβ T- IEL and the WT LN TCRαβ CD8αβ T cells was only 0.85, the lowest value in all the comparisons (*Figure 1f*). In contrast, the proteomes of induced T-IEL and the so-called natural T -IEL populations showed greater similarity with a correlation >0.93 (*Figure 1g and h*), while the correlation between LN T cells from WT to P14 TCR transgenic mice was 0.92 (*Figure 1i*). These analyses indicated that induced T-IEL share a very similar expression profile to natural T-IEL. The comparisons to LN T cells revealed that even LN T cells derived from two different strains of mice were much more similar to each other than to induced T-IEL.

To further explore similarities and differences between induced T-IEL and LN T cells, we focussed on the TCRαβ CD8αβ T-IEL and the WT LN TCRαβ CD8αβ T cells (LN). We first performed a global analysis of the most abundant protein families that represent the top 50 % of the proteome. This overview revealed some similarities and some important proteomic differences between the two cell types. Though the histone content and the glycolytic enzymes looked very similar, there were big differences in proteins related to the ribosomes, the cytoskeleton and the cytotoxic granules (*Figure 2a*). The LN population had nearly double the number of ribosomal proteins, while the T-IEL displayed higher cytoskeletal and cytotoxic proteins. These proteomic differences were not exclusive to TCRαβ CD8αβ T-IEL, as the same pattern was observed within both natural T-IEL subsets. Perhaps, the most striking difference between naïve LN T cells and T-IEL was the expression levels of Granzymes (*Figure 2b*). Granzyme A (GzmA) was expressed at >20 million copies per cell in each of the natural T-IEL subsets, and at 9–10 million molecules per cell in the induced T-IEL population. This was more than double what was previously identified in cytotoxic CD8[+] T cells (*Howden et al., 2019*). Granzyme B (GzmB), which was expressed at ~20 million copies per cell in CTL, was expressed at between 4 and 10 million copies per cell in all three T-IEL subsets. T-IEL also express Granzyme C (GzmC) and K (GzmK), although at <100,000 molecules per cell each (*Figure 2b*), making their general expression of Granzymes either comparable to, or higher than, in vitro-generated CTL. This substantial commitment to Granzyme expression is consistent with the expression of the whole cytotoxic machinery, including perforin and key molecules involved in degranulation (*James et al., 2021*), all of which are either barely detectable, or altogether absent, in the naïve T cells. Thus, these data support the hypothesis that all T-IEL in the gut are geared towards cytotoxic activity.

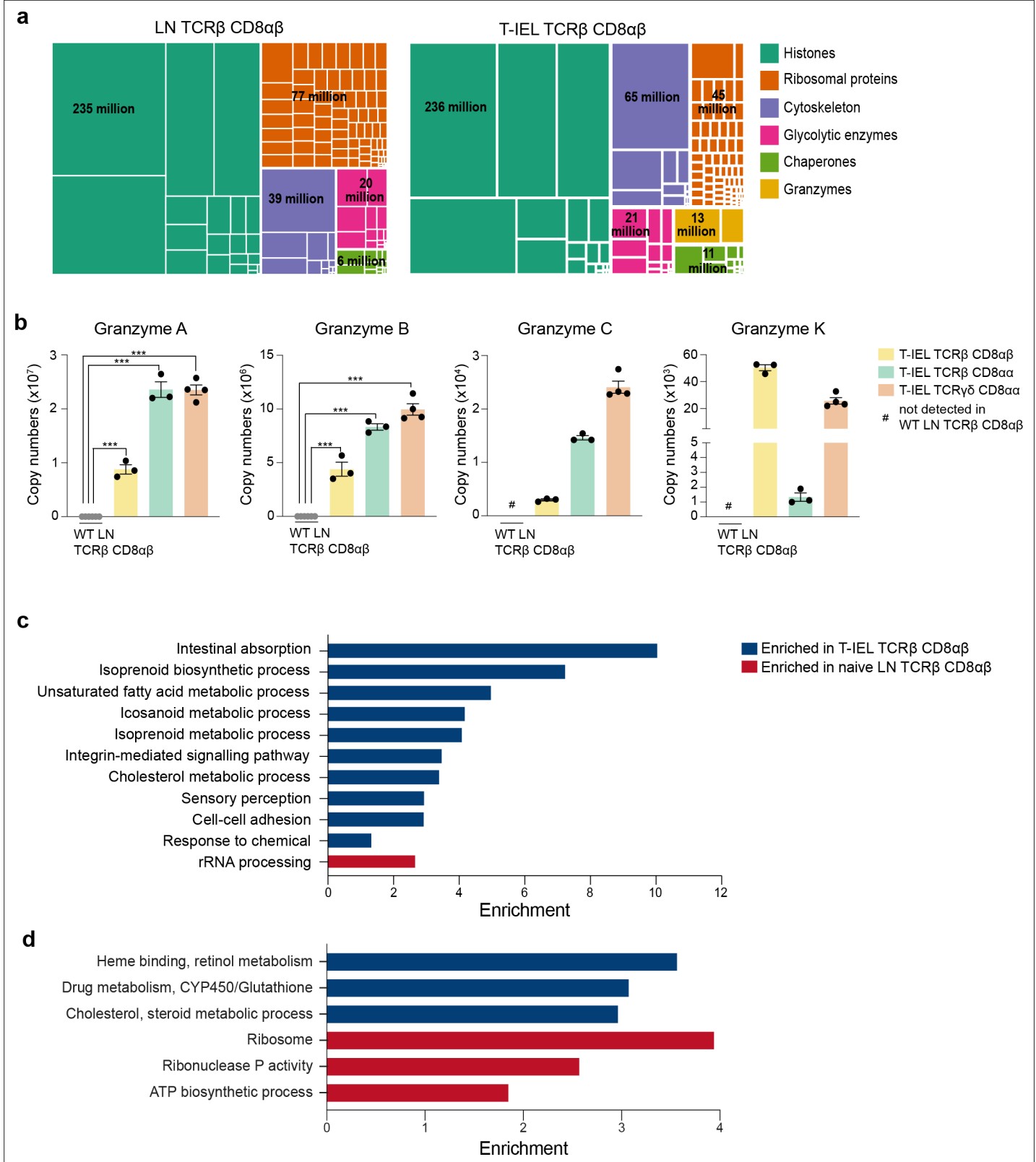

**Figure 2.** Gene ontology analyses of the induced tissue-resident intestinal intraepithelial T lymphocytes (T-IEL) proteome. (**a**) Treemap showing the abundance of proteins classified into histones, ribosomal proteins, cytoskeletal proteins, glycolytic enzymes, chaperones and granzymes across wild-type (WT) lymph node (LN) TCRαβ CD8αβ T cells and TCRαβ CD8αβ T-IEL. Rectangle size is proportional to the median estimated copy numbers. Median copy numbers across all categories are available in **Figure 2—source data 1**. (**b**) Bar plots showing the estimated copy numbers for all

*Figure 2 continued on next page*

*Figure 2 continued*

granzymes across WT LN TCRαβ CD8αβ T cells (n = 6) and all T-IEL (n = 3 or 4). Symbols on the bars represent the biological replicates. The bars show the mean and SEM. The p-values have been calculated on R with LIMMA where ** = p < 0.001 and fold change greater than or equal to the median+1 standard deviation, *** = p < 0.0001 and fold change greater than or equal to the median±1.5 standard deviations. (**c**) Bar plot showing the results of the DAVID functional annotation clustering (false discovery rate [FDR] < 0.05; see Materials and methods for details) enrichment analysis for all proteins exclusive to or significantly increased in expression within TCRαβ CD8αβ T-IEL. (**d**) Bar plot showing the results of the PANTHER GO Biological process (FDR < 0.05; see Materials and methods for details) enrichment analysis for all proteins exclusive to or significantly increased in expression within TCRαβ CD8αβ T-IEL (blue) or within WT TCRαβ CD8αβ T cells (red).

The online version of this article includes the following figure supplement(s) for figure 2:

**Source data 1.** Median copy numbers for the global analysis.

To obtain an unbiased overview of the differences between the induced T-IEL and the LN T cell populations, we performed an overrepresentation analysis (ORA) focussed on gene ontology (GO) terms related to biological processes (*Figure 2c–d*; *Supplementary files 2 and 3*). The data indicated that proteins which were significantly increased in expression in induced T-IEL were highly enriched in cholesterol and lipid metabolism, intestinal absorption and xenobiotic metabolism, and processes involving cytoskeletal proteins, such as cell-cell adhesion and integrin-mediated signalling. Conversely, proteins which were significantly higher expressed in LN T cells were enriched for terms relating to ribosomal proteins and ribonuclease P activity.

## Downregulation of protein synthesis in T-IEL

Based on the results obtained from the ORA, we next focussed on the protein machinery involved with the ribosomes and protein synthesis. A comparison of the total estimated copy numbers for ribosomal proteins indicated that LN T cells express almost double the amount expressed in any of the T-IEL subsets (*Figure 3a*). This was true for both cytoplasmic and mitochondrial ribosomal proteins, with the latter being the most reduced in T-IEL, compared to LN T cells. The decreased expression of ribosomal proteins in T-IEL was mirrored by the decreased expression of RNA polymerases I (Pol1) and III (Pol3), which transcribe, respectively, ribosomal RNA and transfer RNA (*Figure 3b*). For the subunits of both the Pol1 and Pol3 complexes, the median fold reduction in T-IEL was greater than fivefold when compared to LN T cells (*Figure 3—figure supplement 1*). Strikingly, the subunits specific for RNA polymerase II (Pol2), which transcribes protein-coding genes, did not display a reduction in median expression levels. These data suggest that while ribosomal expression is reduced, mRNA pools could potentially still be maintained in T-IEL.

To maintain protein synthesis, consistent uptake of amino acids is generally required. However, our data show that T-IEL express low levels (<2000 copies per cell) of three key amino acid transporters, that is, SLC1A5, SLC7A5, and SLC38A2 (*Figure 3c*), all of which are highly upregulated upon T cell activation with SLC7A5 being expressed at >400,000 copies in effector T cells (*Howden et al., 2019*). SLC7A5 expression levels have been reported to directly control the expression of ribosomal proteins and other important translation machinery components (*Marchingo et al., 2020*; *Sinclair et al., 2019*). The very low levels of amino acid transporters detected in T-IEL is therefore expected to limit protein synthesis. Furthermore, despite having low levels of amino acid transporters, enzymes involved in amino acid catabolism, such as arginase-2 (ARG2) and alanine aminotransferase (glutamic-pyruvic transaminase [GPT]), are highly expressed in T-IEL, also suggesting reduced protein synthesis in T-IEL (*Figure 3d*). Interestingly, high expression of ARG2 was also accompanied by upregulated expression of other enzymes from the urea cycle (*Figure 3—figure supplement 1*). Finally, it is also notable that T-IEL express significantly higher levels of PKR-like Endoplasmic Reticulum Kinase (PERK) than LN T cells. PERK functions as a global protein synthesis inhibitor, either in the presence of unfolded proteins or upon low amino acid availability (*Figure 3e*). We therefore decided to measure the protein synthesis rates in T-IEL and LN T cells by *O*-propargyl puromycin (OPP) incorporation into nascent peptide chains and compared with cycloheximide (CHX)-treated controls. The data from the OPP assay highlighted almost undetectable levels of protein translation in all three T-IEL subsets (*Figure 3f*), which correlated well with the reduced ribosomal content, low expression of amino acid transporters, and high catabolic enzymes identified within the proteomes of T-IEL. In contrast, LN T cells contain more actively translating ribosomes than T-IEL, providing orthogonal validation of the proteomic data. It should be noted that naïve T cells have been reported to have low protein synthesis

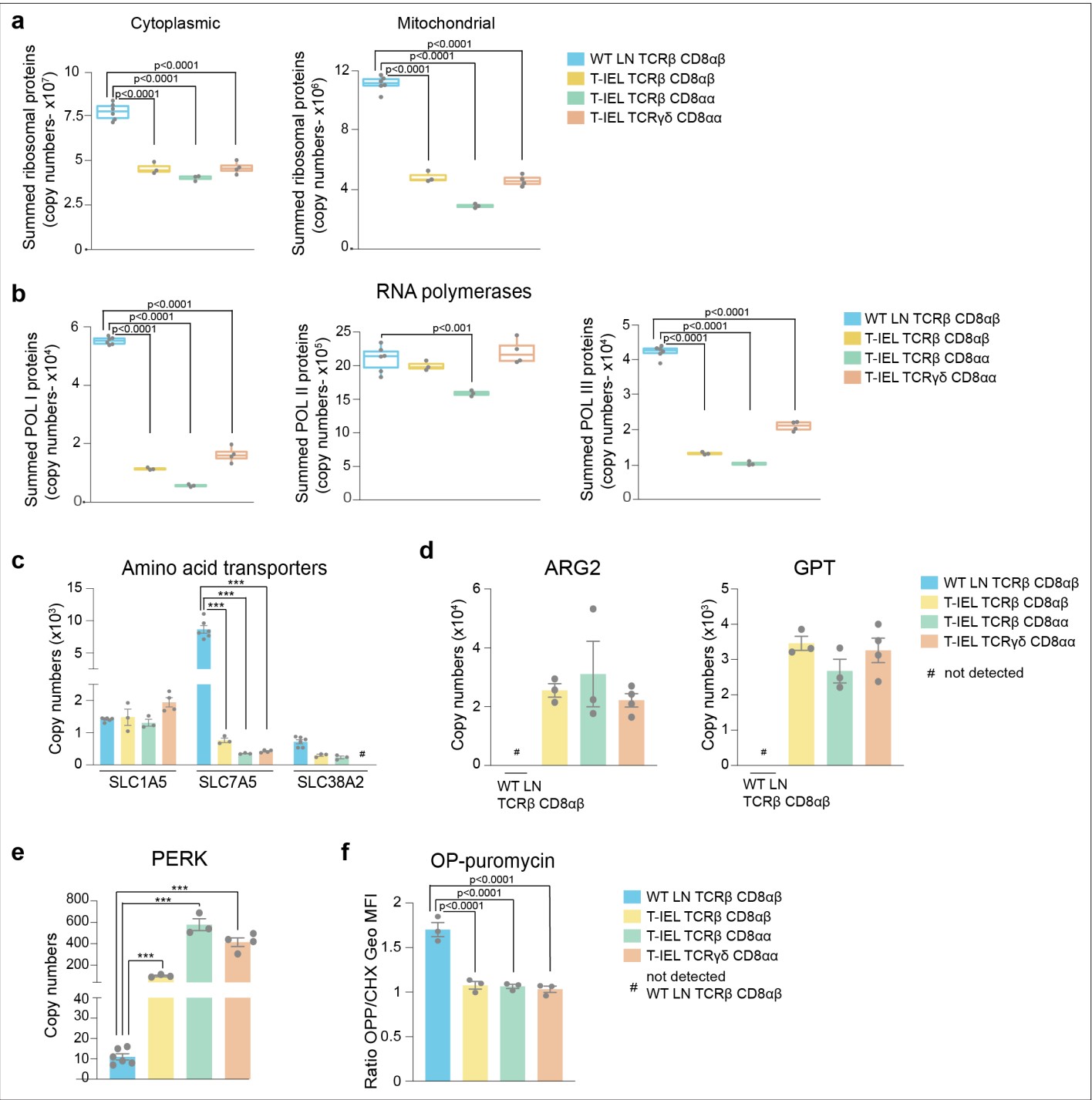

**Figure 3.** Downregulation of protein synthesis in tissue-resident intestinal intraepithelial T lymphocytes (T-IEL). (**a**) Box plots showing the estimated total cytoplasmic (left) and mitochondrial (right) ribosomal protein copies for lymph node (LN) TCRβ CD8αβ T cells and all T-IEL subsets. The sums of all copy numbers are available in *Figure 3—source data 1*. (**b**) Box plots showing the estimated summed total protein copies for the protein subunits that are exclusive to RNA polymerases I, II, and III, respectively, for LN TCRβ CD8αβ T cells and all T-IEL subsets. The sum of all copy numbers are available in *Figure 3—source data 1*. (**c**) Bar plots showing the estimated protein copy numbers of the amino acid transporters, SLC7A5 and SLC38A2, for wild-type (WT) LN TCRβ CD8αβ and all three subsets of T-IEL. (**d**) Bar plots showing the estimated protein copy numbers of arginase-2 (ARG2; left) and alanine aminotransferase (glutamic-pyruvic transaminase [GPT]; right) for WT LN TCRβ CD8αβ and all three subsets of T-IEL. (**e**) Bar plots showing the estimated protein copy numbers of PKR-like endoplasmic reticulum kinase (PERK) for WT LN TCRβ CD8αβ and all three T-IEL subsets. (**f**) Bar plots showing the *O*-propargyl puromycin (OPP) incorporation (n = 3 biological replicates) in ex vivo WT LN TCRβ CD8αβ and T-IEL. As a negative control, OPP incorporation was inhibited by cycloheximide (CHX) pre-treatment. OPP incorporation was assessed by flow cytometry 15 min after administration. Bar graph

*Figure 3 continued on next page*

*Figure 3 continued*

represents the geometric MFI of the OPP-AlexaFluor 647 in each T cell subsets normalised to the geometric MFI of the CHX pre-treated T cells. The p-values were calculated using ordinary one-way ANOVA with Dunnett's multiple comparisons. For all box plots, the bottom and top hinges represent the first and third quartiles. The top whisker extends from the hinge to the largest value no further than 1.5× IQR from the hinge; the bottom whisker extends from the hinge to the smallest value at most 1.5× IQR of the hinge. All bar plots show the mean and SEM. Symbols on the bars represent the biological replicates. The proteomics data displayed on the plots include WT TCRαβ CD8αβ T cells (n = 6 biological replicates), TCRαβ CD8αβ T-IEL and TCRαβ CD8αα T-IEL (n = 3 biological replicates), and TCRγδ CD8αα T-IEL (n = 4 biological replicates). The p-values for individual proteins (c,d,e) were calculated in R with LIMMA, where ** = p < 0.001 and fold change greater than or equal to the median+1 standard deviation, *** = p < 0.0001 and fold change greater than or equal to the median±1.5 standard deviations, and in (a,b) in R with Welch's t-test.

The online version of this article includes the following figure supplement(s) for figure 3:

**Source data 1.** Sum of median copy numbers for the cytoplasmic and mitochondrial ribosomes.

**Figure supplement 1.** Selective regulation of RNA polymerases and the urea cycle in T-IEL.

rates (*Wolf et al., 2020*), however, our data indicate even lower synthesis rates in T-IEL. Thus, multiple mechanisms appear to be active in T-IEL to keep protein synthesis at a minimum.

## T-IEL have a unique metabolic profile

Recent studies have shown a direct correlation between metabolic activity and the rates of protein synthesis in T cells (*Argüello et al., 2020*). The very low levels of protein synthesis in all three T-IEL subsets therefore prompted us to further explore the bioenergetic profile of T-IEL. Globally, we did not find any major differences in the proportion of the T-IEL proteomes dedicated to the major metabolic pathways compared to naïve T cells (*Figure 4a*). We did however find that all three T-IEL subsets express substantial levels of the GLUT2 (~5000 copies) and GLUT3 (~35,000 copies), both facilitative glucose transporters (*Figure 4b*). GLUT2 is normally found in intestinal and other epithelial cells, not in immune cells, and is a low affinity bidirectional glucose transporter. GLUT3 is a high affinity glucose transporter that is thought to be particularly important in CD8 T cell activation (*Geltink et al., 2018*). Glucose can be utilised in T cells through either glycolysis or oxidative phosphorylation (OXPHOS) and the tricarboxylic acid (TCA) cycle that also provides biosynthetic intermediates (*Ma et al., 2019*). We therefore examined the expression of proteins involved in these pathways in T-IEL. We find that T-IEL express most of the proteins of the glycolytic and TCA pathways at similar levels to naïve T cells (*Figure 4c*). With one exception being the lactate transporters, SLC16A1 and SLC16A3, which even though they are significantly higher than in naïve T cells, are still expressed at very low levels, indicating a low glycolytic potential within these cells. Furthermore, T-IEL have also been shown to have comparably low OXPHOS potential as in naïve T cells (*Konjar et al., 2018*). Thus, the function of the glucose being taken up through the T-IEL glucose transporters remains unclear.

We also examined the mitochondrial protein content of T-IEL. The total mitochondrial protein content appeared to be significantly reduced, however, all the components of the electron transport chain (ETC) were expressed at similar levels in all T-IEL, as in LN T cells (*Figure 4d*). These data suggest that T-IEL mitochondria have similar respiratory capacity to naïve T cells. Naïve T cells use OXPHOS and fatty acid oxidation (FAO) to maintain their cellular functions. Therefore, we assessed FAO enzyme expression in T-IEL and found this was also largely similar to naïve T cells (*Figure 4c*). Interestingly, some proteins involved in peroxisomal FAO, including the transporter ABCD4, the key peroxisomal beta-oxidation enzymes acyl-CoA oxidase ACOX1, and carnitine *O*-acetyltransferase, were more highly expressed in T-IEL than in naïve T cells. Peroxisomal FAO produces acetyl CoA, which can be used within the TCA cycle, and NADH, which can be utilised in the ETC, to contribute to energy production. NADH produced during FAO and OXPHOS needs to be transported into the mitochondria through a redox shuttle, and in this context, we find that the glycerol-3-phosphate shuttle is only expressed in T-IEL (*Figure 4e*). Put together, these data suggest that peroxisomes may be a source of fuel to support the low levels of energy produced in T-IEL and indicate key differences in the metabolic pathways active in T-IEL.

## T-IEL have increased lipid biosynthesis and cholesterol metabolism

Our data would seem to indicate that T-IEL have low bioenergetic production and requirements. However, functional annotation of proteins enriched in induced T-IEL indicate overrepresentation of cholesterol and steroid metabolism pathways, and the metabolism of chemicals and inorganic

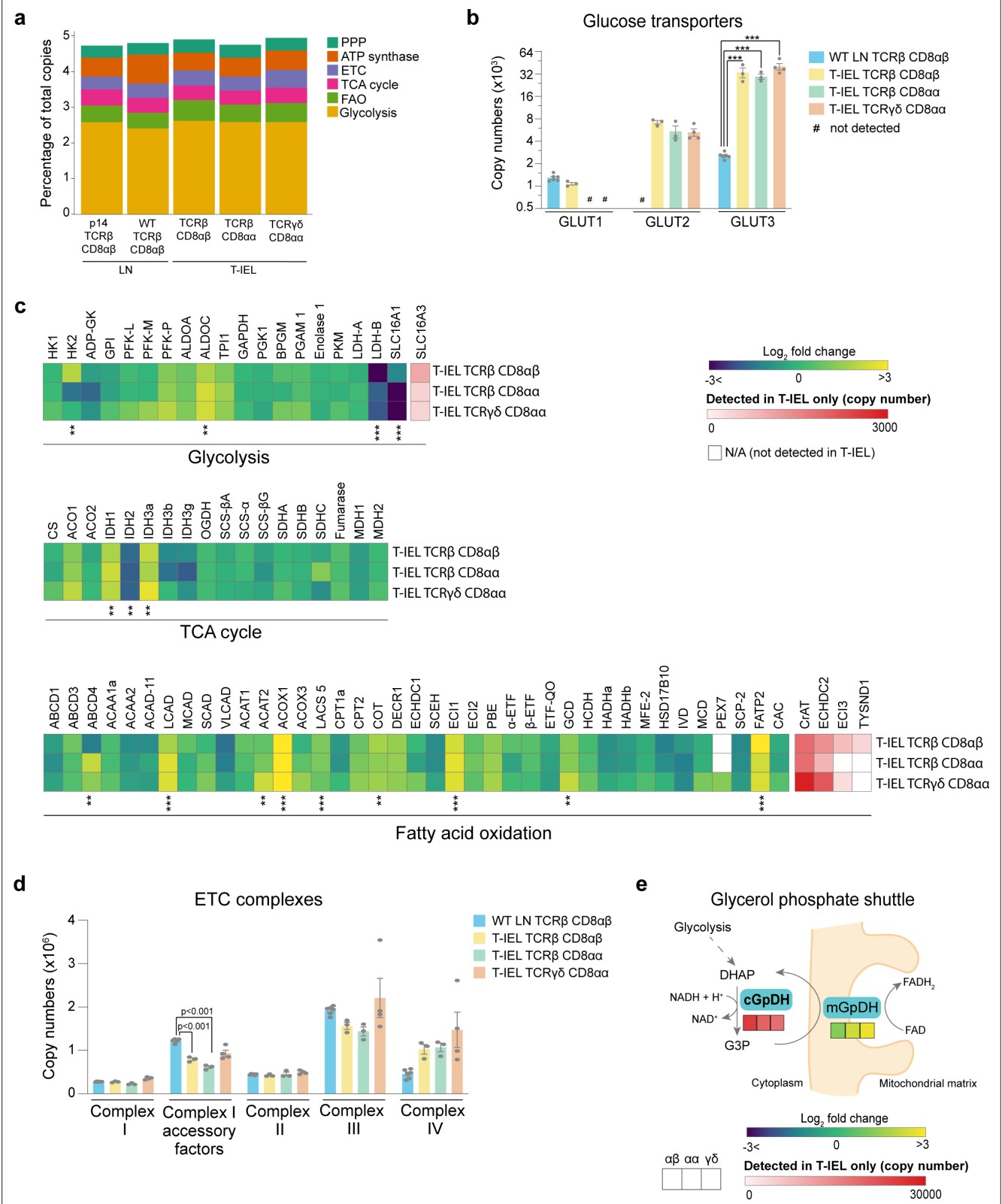

**Figure 4.** Metabolic profiling of the tissue-resident intestinal intraepithelial T lymphocytes (T-IEL) proteome. (**a**) Stacked bar plots comparing the proportional representation of metabolic pathways in lymph node (LN) TCRβ CD8αβ and all T-IEL subsets. (**b**) Bar plots showing the estimated protein copy numbers of the glucose transporters, GLUT1, GLUT2, and GLUT3, for wild-type (WT) LN TCRβ CD8αβ and all three subsets of T-IEL. (**c**) Heatmaps displaying the Log$_2$ fold change (T-IEL/ LN CD8 T cells) for all proteins involved in glycolysis, tricarboxylic acid (TCA) cycle, and fatty acid oxidation

*Figure 4 continued on next page*

*Figure 4 continued*

(FAO). (**d**) Bar plots showing the sum of the estimated protein copy numbers of the electron transport chain (ETC) components, for WT LN TCRβ CD8αβ and all three subsets of T-IEL. Sum of the copy numbers across all ETC complexes are available in **Figure 4—source data 1**. (**e**) Schematic representation of the glycerol-3-phosphate shuttle with heatmaps showing protein expression of cytosolic glycerol-3-phosphate dehydrogenase (cGpDH) and the $Log_2$ fold change of mitochondrial glycerol-3-phosphate dehydrogenase (mGpDH) (T-IEL/LN CD8 T cells) in, from left to right, T-IEL TCRβ CD8αβ, T-IEL TCRβ CD8αα, and T-IEL TCRγδ CD8αα. All bar plots show the mean and SEM. Symbols on the bars represent the biological replicates. The proteomics data displayed on the plots include WT TCRαβ CD8αβ T cells (n = 6 biological replicates), TCRαβ CD8αβ T-IEL and TCRαβ CD8αα T-IEL (n = 3 biological replicates), and TCRγδ CD8αα T-IEL (n = 4 biological replicates). The p-values for individual proteins (**b–c**) were calculated in R with LIMMA, where ** = p < 0.001 and fold change greater than or equal to the median+1 standard deviation, *** = p < 0.0001 and fold change greater than or equal to the median±1.5 standard deviations, and in (**d**) in R with Welch's t-test. For full protein names, see **Supplementary file 4**.

The online version of this article includes the following figure supplement(s) for figure 4:

**Source data 1.** Sum of median copy numbers for all the electron transport chain (ETC) complexes.

compounds (**Figure 2c**). T-IEL are highly enriched in proteins involved in xenobiotic metabolism, including members of the UDP glucuronosyl transferase family, glutathione *S*-transferase (GST), and cytochrome P450 enzymes (**Supplementary file 1**). Detailed examination of the cholesterol biosynthetic pathway indicates almost all the enzymes are expressed highly in all T-IEL, as compared to LN T cells (**Figure 5a**). This pathway is controlled by the master regulator sterol-regulatory element binding protein 2 (**Madison, 2016**), which the data showed is exclusively expressed within the three T-IEL populations (**Figure 5b**). We therefore measured cholesterol content in T-IEL and found that indeed all three subsets have greater than 2.5-fold more cholesterol than naïve LN CD8 T cells (**Figure 5c**).

T-IEL also express the fatty acid transport proteins (FATP2(*Slc27a2*) and FATP4 (*Slc27a4*)), which are necessary for uptake and transport of long chain fatty acids, as well as fatty acid binding proteins (FABP1, -2, -5, and -6), which also contribute to uptake and transport of fatty acids to the endoplasmic reticulum (**Figure 5d and e**). In addition to the intestinal specific family member, FABP2 (>350,000 copies/cell), the liver FABP, FABP1 (>200,000 copies/cell), which is highly expressed in the proximal intestine, and the ileal FABP, FABP6, or gastrotropin (>10,000 copies/cell), are all also highly expressed in all three T-IEL subsets (**Figure 5e**). It is interesting to note that FABP5, which was previously identified as being expressed in skin T$_{RM}$ cells, but not in intestinal T$_{RM}$ at the mRNA level (**Frizzell et al., 2020**), was detected at >200,000 molecules per cell in all three T-IEL subsets. Skin T$_{RM}$ appear to use increased exogenous fatty acids uptake to feed into mitochondrial FAO, thus supporting their maintenance and survival (**Pan et al., 2017**). However, carnitine *O*-palmitoyl transferase (CPT1A), the rate-limiting enzyme for mitochondrial FAO of long chain fatty acids is expressed at lower levels in T-IEL compared to naïve LN T cells (**Figure 4c**). This suggests that the highly increased lipid transporter expression in T-IEL is not solely used to drive FAO.

T-IEL are also enriched in proteins involved in the two major pathways of triacylglycerol (TAG or triglyceride) synthesis expressed in the intestine (**Figure 5f**; **Yen et al., 2015**). TAG is hydrophobic and is either stored transiently in the cytosol in lipid droplets or assembled and secreted from enterocytes in apolipoprotein B (ApoB)-containing chylomicrons, or lipoproteins that also contain cholesterol and cholesteryl esters. Surprisingly, T-IEL also express high levels of a key cholesterol esterification enzyme, Acyl CoA:cholesterol acyl transferase 2, ACAT-2 (*Soat2*), which is thought to be specifically expressed in enterocytes (**Pan and Hussain, 2012**). Esterification of cholesterol increases its hydrophobicity for efficient packaging into lipoproteins. We therefore also explored the expression of enzymes involved in lipoprotein assembly. Lipoprotein assembly involves the packaging of TAG and cholesteryl esters by the microsomal triglyceride transfer protein (MTP, *Mttp*) into ApoB-lipid conjugates, followed by export out of the cells by the core protein complex II (COPII) (**Hussain et al., 2012**). MTP was highly expressed in T-IEL with over 80,000 copies per cell, while <100 copies were identified in LN T cells. Similarly, ApoB and the GTPase SAR1b, a key component of the COPII complex, were also expressed in T-IEL at higher copies than in LN T cells (**Figure 5f**). Together, these data suggest that T-IEL also take up and metabolise fatty acids and cholesterol, and further, have the capacity to package these lipids into lipid droplets and potentially even transport them out of the cells.

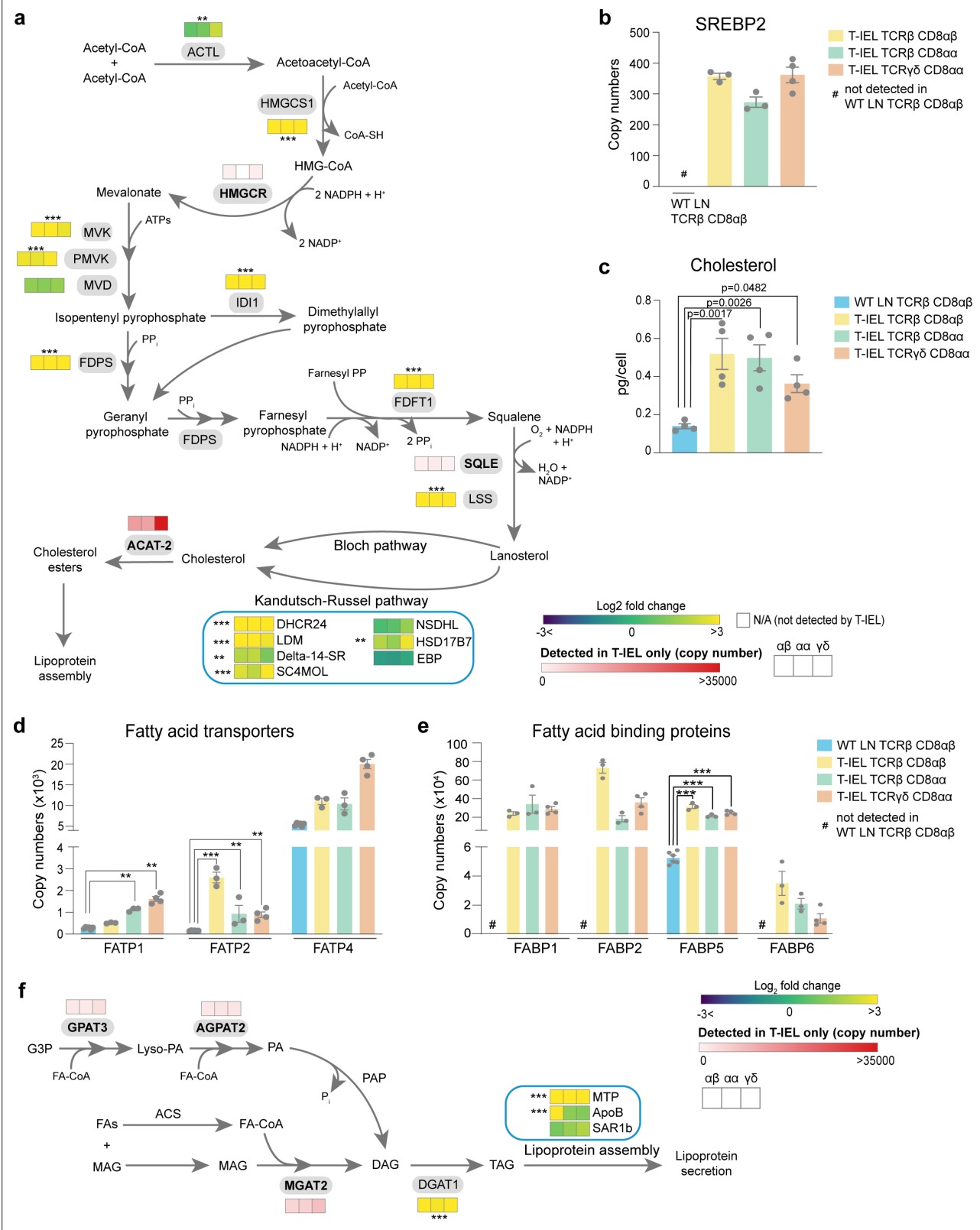

**Figure 5.** Tissue-resident intestinal intraepithelial T lymphocytes (T-IEL) have enhanced cholesterol and lipid metabolism. (**a**) Schematic representation of proteins involved of the cholesterol biosynthetic pathway. Heatmap squares represent the Log$_2$ fold change (T-IEL/lymph node [LN] CD8 T cells) in, from left to right, T-IEL TCRβ CD8αβ, T-IEL TCRβ CD8αα, and T-IEL TCRγδ CD8αα. Proteins expressed only by T-IEL are highlighted by red squares, representing the mean estimated protein copy numbers (from at least three biological replicates). (**b**) Bar plot showing the estimated protein copy

*Figure 5 continued on next page*

*Figure 5 continued*

number of SREBP2 for wild-type (WT) LN TCRβ CD8αβ and all three subsets of T-IEL. (**c**) Bar plot showing a comparison of total cellular cholesterol content in sorted WT LN TCRβ CD8αβ and all three subsets of T-IEL (n = 4 biological replicates). The p-values calculated using ordinary one-way ANOVA with Dunnett's multiple comparison test. Data for the total cholesterol content are available in *Figure 5—source data 1*. (**d**) Bar plots showing the estimated protein copy numbers of the fatty acid transporters FATP1, FATP2, and FATP4 for WT LN TCRβ CD8αβ and all three subsets of T-IEL. (**e**) Bar plots showing the estimated protein copy numbers of the fatty acid binding proteins FABP1, FABP2, FABP5, and FABP6 for WT LN TCRβ CD8αβ and all three subsets of T-IEL. (**f**) Schematic representation of the triacylglycerol synthesis pathways and lipoprotein assembly. Coloured squares represent the Log$_2$ fold change (T-IEL/LN CD8 T cells) in, from left to right, T-IEL TCRβ CD8αβ, T-IEL TCRβ CD8αα, and T-IEL TCRγδ CD8αα. Proteins expressed only by T-IEL are highlighted by red squares, representing estimated protein copy numbers (mean from at least three biological replicates). All bar plots show the mean and SEM. Symbols on the bars represent the biological replicates. The proteomics data displayed on the plots include WT TCRαβ CD8αβ T cells (n = 6 biological replicates), TCRαβ CD8αβ T-IEL and TCRαβ CD8αα T-IEL (n = 3 biological replicates), and TCRγδ CD8αα T-IEL (n = 4 biological replicates). The p-values for individual proteins (**a,b,d,e,f**) were calculated in R with LIMMA, where ** = p < 0.001 and fold change greater than or equal to the median+1 standard deviation, *** = p < 0.0001 and fold change greater than or equal to the median±1.5 standard deviations. For full protein names, see *Supplementary file 4*.

The online version of this article includes the following figure supplement(s) for figure 5:

**Source data 1.** Total cholesterol content across all populations.

## Intestinal T-IEL proteome contains cell surface receptors for epithelial and neuroimmune interactions

We also explored the expression of proteins uniquely identified in T-IEL and found several proteins involved in cell adhesion, cytoskeleton remodelling, and integrin signalling. Strikingly, all T-IEL subsets expressed numerous epithelial cell adhesion molecules and integrins which are not found on naïve LN T cells (*Figure 6a*). Although these results are consistent with the localisation of T-IEL within the gut epithelial layer, we were surprised to find T-IEL proteomes also contained many tight junction, adherens junction, and desmosome-associated proteins, which are normally expressed on intestinal epithelial cells, such as E-cadherin (E-Cad), ZO-2, desmoplakin, villin-1, and JAM-A (F11R) (*Figure 6a and b*). These proteins could potentially be contaminants from epithelial cells in the sample preparation, however, E-Cad, occludin, and EpCAM have been detected both at the RNA and protein level in T-IEL (*Figure 6—figure supplement 1*, *Inagaki-Ohara et al., 2005*; *Nochi et al., 2004*). Moreover, immunofluorescence imaging and flow cytometry confirmed expression of ZO-2, E-Cad, and EpCAM in T-IEL, suggesting that T-IEL could use these molecules to navigate the tissue environment (*Figure 6c*, *Figure 6—figure supplement 1*). Conversely, endothelial cell adhesion molecules such as PECAM-1 and L-selectin, that facilitate T cell migration into secondary lymphoid organs, were highly expressed on naïve T cells, but not on T-IEL, as befits their tissue-resident status (*Figure 6d*).

T-IEL proteomes also suggest that T-IEL could be communicating with the enteric nervous system. TCRγδ CD8αα T-IEL express two neural cell adhesion molecules, NCAM1 (CD171) and NrCAM, both implicated in homophilic adhesion and in axonal growth and guidance. Furthermore, two neuropeptide receptors, GPR171 and VIPR2, were also identified in T-IEL proteomes (*Figure 6b*). BigLEN and vasoactive intestinal peptide (VIP) bind to GPR171 and VIPR1/VIPR2, respectively (*Delgado et al., 2004*; *Gomes et al., 2013*). BigLEN and VIP are neuropeptides with multiple physiological effects, including gut motility, nutrient absorption, food intake regulation, and immune responses (*Yoo and Mazmanian, 2017*). VIPR2 expression on intestinal innate lymphoid cells was shown to regulate their immune response (*Seillet et al., 2020*; *Talbot et al., 2020*). In addition, we also found that T-IEL express GLP1R and GLP2R (*Figure 6b*), receptors for the glucagon-like peptides 1 and 2 (GLP1 and GLP2), which are intestinal peptides involved in regulating appetite and satiety. Both of these receptors were previously mainly found on enteroendocrine cells and enteric neurons. However, recently GLP1R expression on T-IEL was shown to contribute to metabolic syndrome development in mice (*He et al., 2019*; *Yusta et al., 2015*). Together, these data suggest that T-IEL may be involved in regulating immune responses and potentially also metabolic responses to food intake through their communication with epithelial cells.

## T-IEL share a common signature with exhausted T cells

T-IEL also express many signalling receptors that are absent on naïve T cells and that potentially regulate their poised activated state (*Vandereyken et al., 2020*). The proteomic analyses here confirmed that all three T-IEL subsets express many inhibitory receptors, including LAG-3, CD200R1, CD244

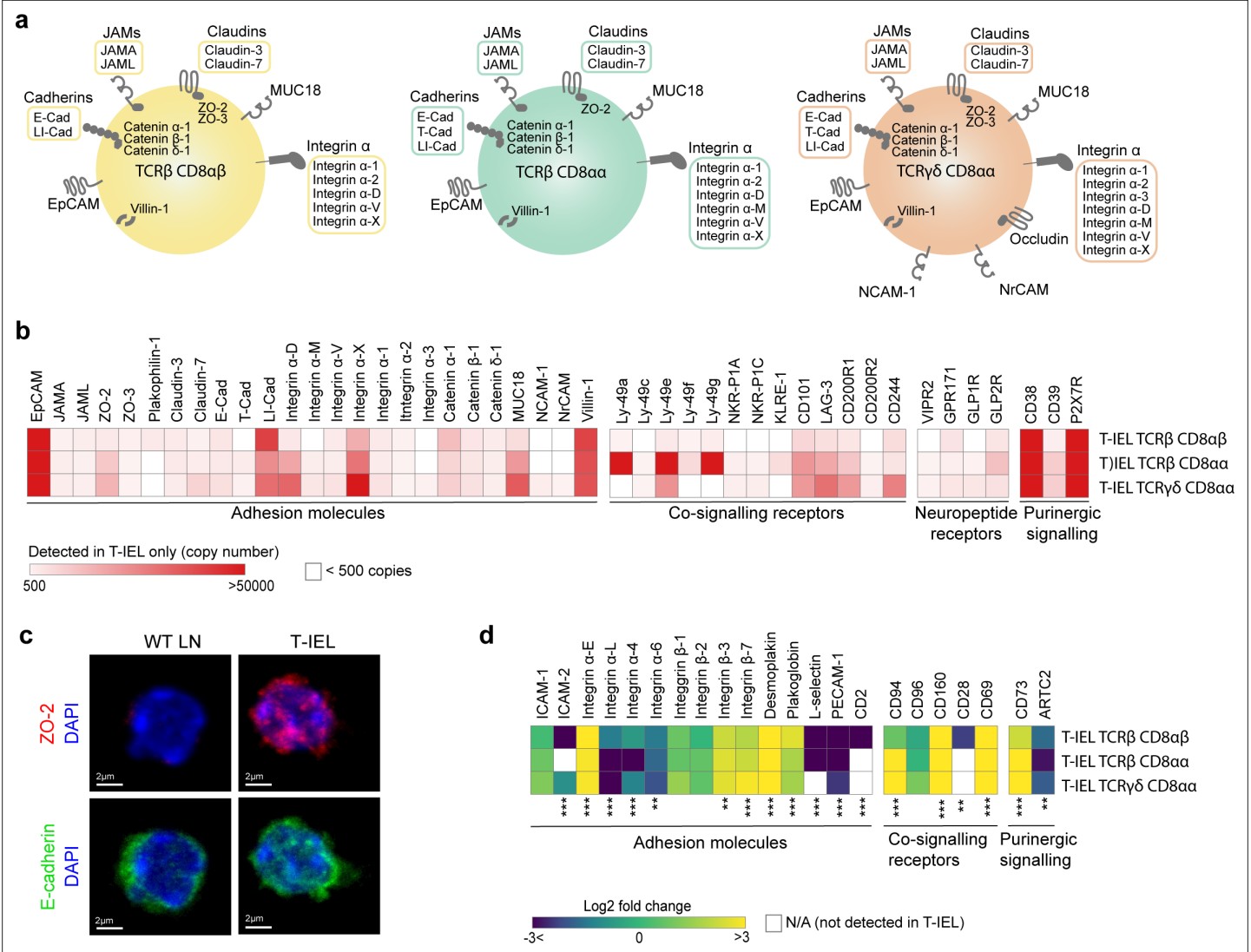

**Figure 6.** Cell surface proteins expressed on tissue-resident intestinal intraepithelial T lymphocytes (T-IEL). (**a**) Schematic representation of proteins involved in cell-cell adhesion that are only expressed by the different T-IEL subsets. (**b**) Heatmaps displaying the estimated protein copy numbers of adhesion molecules, co-signalling receptors, neuropeptide receptors, and purinergic receptors expressed only by T-IEL. Data represent the mean of at least three biological replicates. (**c**) Purified lymph node (LN) CD8 T cells (left) and isolated T-IEL (right) were immunostained for ZO-2 (top, red), E-cadherin (bottom, green), and CD8α (not shown) and counterstained with DAPI to mark the nuclei (blue). Representative (of two independent experiments) maximal intensity projections of confocal sections spanning the entire cell thickness of selected CD8+ cells of each type are shown. Size bars = 2 μm. See also *Figure 6—figure supplement 1*. (**d**) Heatmap displaying $Log_2$ fold change (T-IEL/ LN CD8 T) cells of adhesion molecules, co-signalling receptors, and purinergic receptors. The proteomics data displayed on the plots show the mean values and were calculated from wild-type (WT) TCRαβ CD8αβ T cells (n = 6 biological replicates), TCRαβ CD8αβ T-IEL and TCRαβ CD8αα T-IEL (n = 3 biological replicates), and TCRγδ CD8α T-IEL (n = 4 biological replicates). The p-values were calculated in R with LIMMA, where ** = p < 0.001 and fold change greater than or equal to the median+1 standard deviation, *** = p < 0.0001 and fold change greater than or equal to the median±1.5 standard deviations.

The online version of this article includes the following source data and figure supplement(s) for figure 6:

**Figure supplement 1.** Expression of epithelial proteins in tissue-resident intestinal intraepithelial T lymphocytes (T-IEL).

**Figure supplement 1—source data 1.**

and NK receptors, such as members of the Ly49 family, but also showed that a wider range of these inhibitory receptors are found on innate T-IEL compared to induced T-IEL (*Figure 6b*). Furthermore, T-IEL, regardless of ontogeny, uniformly expressed CD38 and CD73 (*Nt5e*) (*Figure 6b and d*). Indeed, co-expression of CD38 and CD73 is seen to provide a better marker for identifying T-IEL than CD103 expression (*Figure 7a and b*). These receptors are tightly linked to purinergic signalling through their

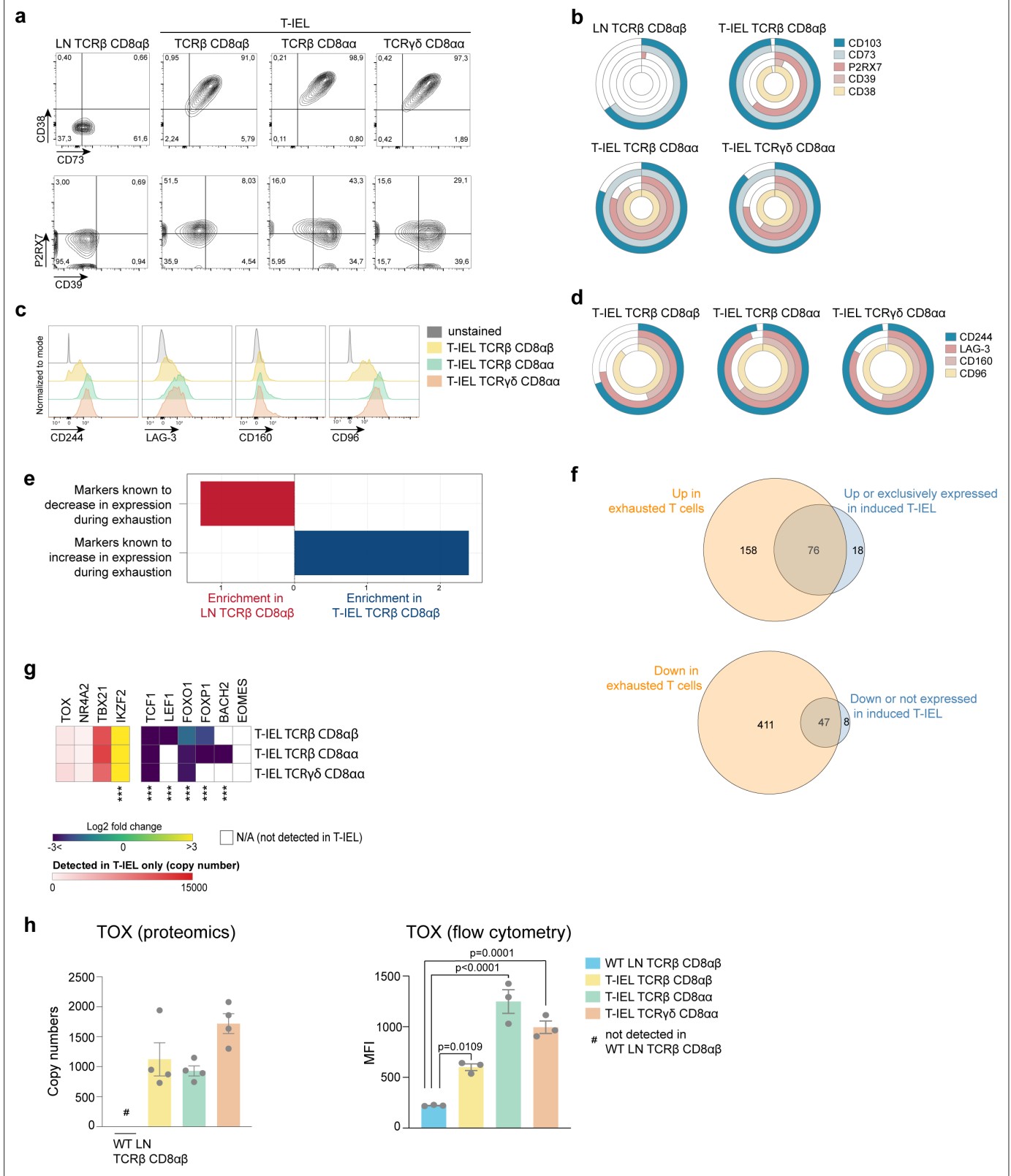

**Figure 7.** Tissue-resident intestinal intraepithelial T lymphocytes (T-IEL) share similarities with exhausted T cells. (**a**) Flow cytometry dot plots comparing the expression of purinergic receptors (CD38/CD73, top and P2RX7 /CD39, bottom) in wild-type (WT) lymph node (LN) TCRβ CD8αβ and all three subsets of T-IEL. (**b**) Stacked doughnut plot showing the percentages of cells from LN CD8 T cells and T-IEL expressing the indicated purinergic receptors, quantified by flow cytometry (n = 1 biological replicate). (**c**) Flow cytometric histograms comparing the protein expression of the exhaustion

*Figure 7 continued on next page*

*Figure 7 continued*

markers, CD244, LAG-3, CD160, and CD96 in all three subsets of T-IEL. (**d**) Stacked doughnut plot showing the percentage of cell from LN CD8 T cells and T-IEL expressing the indicated exhaustion markers quantified by flow cytometry (n = 4 biological replicates). (**e**) Bar plot showing the result of the T cells exhaustion overrepresentation analyses in LN TCRβ CD8αβ and in TCRβ CD8αβ T-IEL. (**f**) Venn diagrams showing the commonality of proteins upregulated (top) and downregulated (bottom) during exhaustion and in TCRβ CD8αβ T-IEL. (**g**) Heatmap displaying the Log$_2$ fold change (T-IEL/LN CD8 T cells) of transcription factors associated with exhaustion in T cells. (**h**) Bar plots showing the estimated protein copy number of TOX (left) and flow cytometry derived MFI for TOX (right) for WT LN TCRβ CD8αβ and all three subsets of T-IEL. The proteomics data displayed on the plots include WT TCRαβ CD8αβ T cells (n = 6 biological replicates), TCRαβ CD8αβ T-IEL and TCRαβ CD8αα T-IEL (n = 3 biological replicates), and TCRγδ CD8αα T-IEL (n = 4 biological replicates). The flow cytometry was performed on three biological replicates, representative of two independent experiments. The p-values for individual proteins (**g–h**) were calculated in R with LIMMA where ** = p < 0.001 and fold change greater than or equal to the median+1 standard deviation, *** = p < 0.0001 and fold change greater than or equal to the median±1.5 standard deviations, for the flow cytometry data (**h**) in GraphPad Prism using one-way ANOVA with Dunnett's multiple comparisons test. For full protein names, see *Supplementary file 4*.

The online version of this article includes the following figure supplement(s) for figure 7:

**Source data 1.** Flow cytometry-based percentage of cells expressing exhaustion markers and TOX MFI.

regulation of P2RX7, and as previously found on T$_{RM}$ cells (**Borges da Silva et al., 2018**; **Stark et al., 2018**), P2RX7 and CD39 are also highly expressed on T-IEL, although less uniformly than CD38 and CD73 (**Figure 7a and b**).

CD38 and CD39 have recently been identified as markers of T cell exhaustion, along with expression of PD-1, LAG-3, CD244, CD160 among other inhibitory receptors. As all these molecules are highly expressed on T-IEL (**Figure 7a–d**), with the exception of PD-1, T-IEL appear to share some similarities with exhausted T cells (**Alfei et al., 2019**; **Khan et al., 2019**; **Scott et al., 2019**). An ORA using a database of T cell exhaustion markers confirmed that T-IEL are enriched in markers of exhaustion (**Figure 7e**), with at least 76 proteins that were upregulated in exhausted T cells also being upregulated in T-IEL (**Figure 7f**, top and **Supplementary file 5**). During exhaustion of systemic T cells, several proteins are downregulated. Interestingly, a significant proportion of these downregulated proteins are also downregulated in induced T-IEL (**Figure 7f**, bottom and **Supplementary file 5**). We therefore further examined the expression of transcription factors associated with T cell exhaustion (**Figure 7g**). Indeed, two transcription factors recently identified to be key to imprinting the 'exhausted' T cell phenotype, that is, TOX and NR4A2, were preferentially expressed in all T-IEL, whereas other transcription factors that show reduced expression in exhausted T cells, including TCF1 and LEF1, were also downregulated in T-IEL. We further confirmed expression of TOX in all T-IEL subsets by flow cytometry (**Figure 7h**). However, T-IEL still express high levels of T-bet, as do effector T cells, which most likely helps to maintain expression of cytolytic effector molecules, such as granzymes, while repressing PD-1 expression on T-IEL. Overall, both natural and induced T-IEL appear to have a hybrid phenotype combining features of exhausted T cells and effector T cells, while also bearing unique hallmarks imprinted by the intestinal microenvironment.

## Modifications in the TCR signalosome in T-IEL

Given the connection between T-IEL and exhausted T cells, one key question we wanted to address was whether T-IEL are also unresponsive to TCR stimulation. Indeed, we find that only a small percentage (<10%) of both natural T-IEL subsets are able to respond to TCR cross-linking as measured by induction of phosphorylation of ERK1/2 and S6 ribosomal protein (**Figure 8a and b**). However, induced TCRαβ CD8αβ T-IEL responded even better than LN T cells to TCR stimulation. It has previously been recognised that cross-linking of the TCR on TCRγδ T-IEL does not induce calcium flux and downstream signalling (**Malinarich et al., 2010**; **Wencker et al., 2014**). This reduced TCR signalling capacity has been attributed to chronic TCR signalling in the tissue. However, how TCR signalling is dampened at a mechanistic level has not yet been addressed. We sought to evaluate whether there were changes in the TCR signalosome in T-IEL that were blocking TCR signals, and how conserved it was across the different subsets (**Figure 8c**). Strikingly, several proteins were differentially expressed, not just in TCRγδ T-IEL, but in all T-IEL subsets including induced TCRαβ CD8αβ T-IEL. Quantitative analysis of the immediate TCR signalling elements confirmed previous studies showing exclusive expression of FcεR1γ and LAT2 (NTAL/LAB) on T-IEL, and downregulation of LAT and CD3 ζ as compared to LN T cells (**Figure 8c**). Replacement of the CD3 ζ chain with the FcεR1γ chain reduces the number of immunoreceptor tyrosine-based activation motifs in the TCR. LAT2 is reported to play a dominant

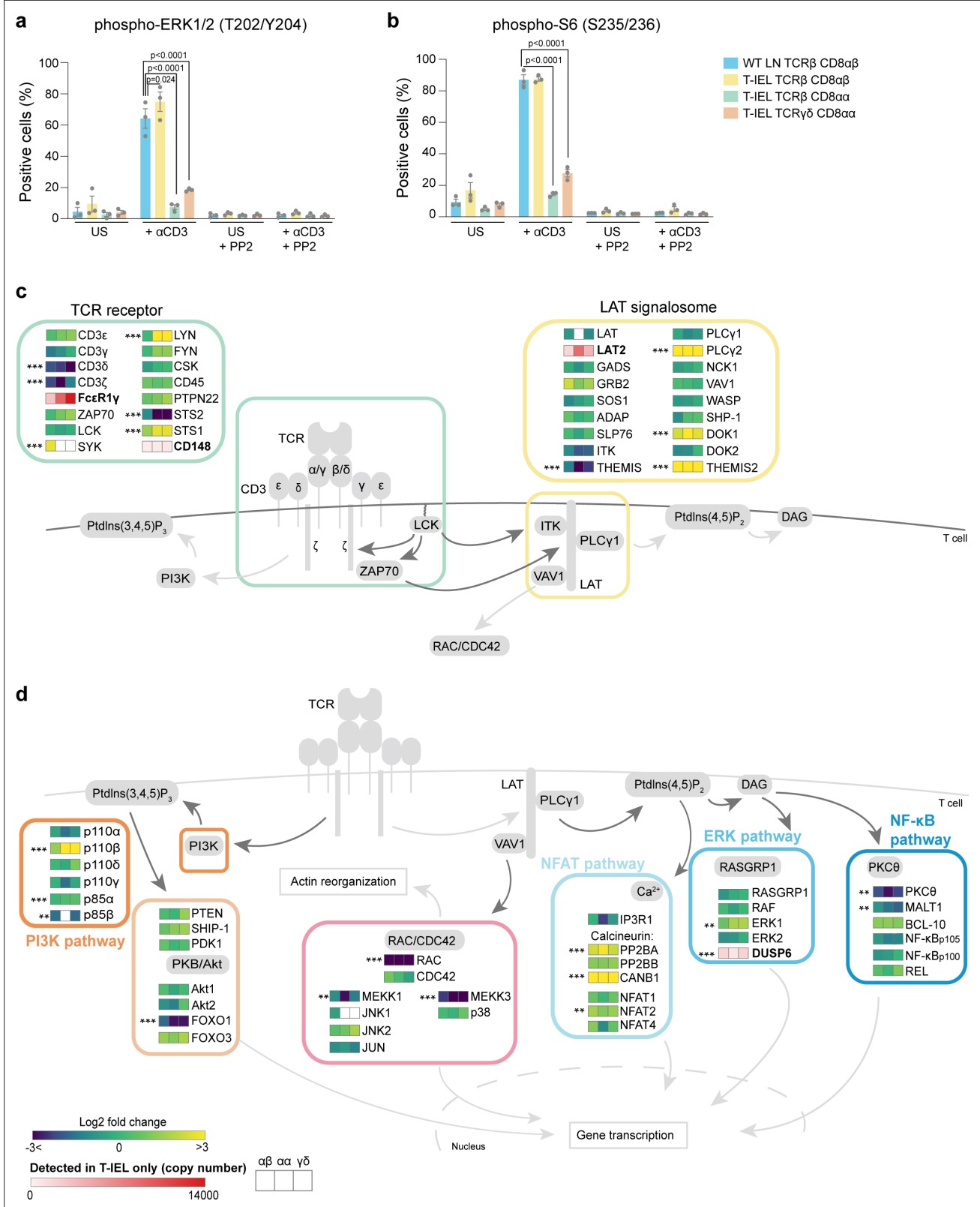

**Figure 8.** Rewiring of the T cell antigen receptor (TCR) signalosome in tissue-resident intestinal intraepithelial T lymphocytes (T-IEL). (**a–b**) Responses of wild-type (WT) lymph node (LN) CD8 T cells and T-IEL to TCR stimulation. Bar plots show the percentage of cells positive for (**a**) phospho-ERK1/2 (T202/Y204) and (**b**) phospho-S6 (S235/236) before and after anti-CD3 stimulation. The Src kinase inhibitor PP2 was added as a control to show that induction of ERK1/2 and S6 phosphorylation was specific. n = 3 biological replicates, p-values were calculated by two-way ANOVA with Dunnett's multiple

*Figure 8 continued on next page*

*Figure 8 continued*

comparisons test. Data are available in *Figure 8—source data 1*. (**c–d**) Schematic representation of the main TCR signalling pathways comparing the expression of selected proteins in T-IEL and LN naïve T cells. (**c**) TCR and LAT signalosome. (**d**) Signalling pathways downstream TCR receptor. All heatmap squares represent the Log₂ fold change (T-IEL/LN CD8 T cells) in, from left to right, T-IEL TCRβ CD8αβ, T-IEL TCRβ CD8αα, and T-IEL TCRγδ CD8αα. Proteins expressed only by T-IEL are highlighted by red squares, representing estimated protein copy numbers (mean from at least three biological replicates). The proteomics data displayed on the plots was calculated from WT TCRαβ CD8αβ T cells (n = 6 biological replicates), TCRαβ CD8αβ T-IEL and TCRαβ CD8αα T-IEL (n = 3 biological replicates), and TCRγδ CD8αα T-IEL (n = 4 biological replicates). The p-values were calculated in R with LIMMA, where ** = p < 0.001 and fold change greater than or equal to the median+1 standard deviation, *** = p < 0.0001 and fold change greater than or equal to the median±1.5 standard deviations. For protein names, see *Supplementary file 4*.

The online version of this article includes the following figure supplement(s) for figure 8:

**Source data 1.** Flow cytometry-based percentage of cells positive for phospho ERK1/2 and phospho-S6.

negative role in TCR signalling by competing with LAT for binding partners, but is unable to couple to PLCγ (*Fuller et al., 2011*).

In addition to LAT and CD3 ζ , several other proteins were differentially expressed (*Figure 8c and d*). Surprisingly, many proteins normally found in B cells and involved in BCR signal transduction were identified as expressed in T-IEL, for example, Lyn, Syk, LAT2, PLCγ2, Themis2, and many of these are also often found in exhausted T cells (*Supplementary file 5*, *Khan et al., 2019*; *Schietinger et al., 2016*). We also noted the expression of several negative regulators of TCR signalling, including STS-1 (Ubash3b) that dephosphorylates Zap70/Syk, CD148 (PTPRJ), and DUSP6, which negatively regulates MAPK signalling (*Gaud et al., 2018*; *Figure 8c and d*). These negative regulators of signalling are also highly expressed in tumour-associated exhausted T cells (*Schietinger et al., 2016*). Conversely, key TCR signalling intermediates, such as protein kinase C θ and Rac, were very poorly expressed. Importantly, many of these changes were not confined to the natural CD8αα T-IEL subsets but were also identified in induced T-IEL.

In summary, these data suggest that the rewiring of the TCR signalosome in T-IEL occurs independently of the developmental pathway through which the three different subsets are derived, and is instead shaped by the intestinal environment. However, as induced T-IEL also express LAT2 and FcεR1γ chain, but still respond to TCR signals, the loss of TCR responsiveness in natural T-IEL cannot be solely attributed to these proteins. Further evaluation of the TCR signalling pathways is necessary to provide an explanation for the loss of TCR responses in natural T-IEL.

## Discussion

Both TCRγδ and TCRαβ CD8αα natural T-IEL have long been considered unconventional T cells, due to their unique developmental pathways and their strict restriction to the intestinal epithelium. In contrast, induced T-IEL that arise from systemic antigen-experienced T cells are considered conventional and more like memory T cells in their ability to respond rapidly to activation signals. Yet, our unbiased analyses clearly show that induced T-IEL share far greater similarity to other intestinal T cell subsets than to the systemic T cells they arise from. The T-IEL signature most strikingly contains several proteins thought to be exclusively, or very highly, expressed by enterocytes. These include cognate proteins involved in mediating adherens and tight junction formation, showing that T-IEL are strongly integrated into the intestinal epithelium, by interactions that extend well beyond the CD103:E-cadherin interaction. T-IEL also share metabolic similarities with enterocytes, including a strong enrichment in proteins required for cholesterol, lipid, and xenobiotic metabolism. Many of these genes are aryl hydrocarbon receptor (AHR) targets (*Stockinger et al., 2014*; *Tanos et al., 2012*), suggesting that one reason why AHR is essential for T-IEL survival (*Li et al., 2011*) is to protect them from toxins and bacterial metabolites in the gut. Other potential indicators of tissue adaptation include expression of the intestine-specific GLUT2 glucose transporter, and the high expression of the urea cycle that could be important for detoxifying the large quantities of ammonia that is present in the intestinal lumen (*Romero-Gomez et al., 2009*). Furthermore, we find that despite having very low energy requirements, T-IEL have a distinct metabolic signature, with high expression of proteins such as GLUT3, glycerol-3-phosphate shuttle, and peroxisomal FAO enzymes. Interestingly, the glycerol phosphate shuttle is normally only expressed in highly glycolytic cells to maintain cellular redox balance by recycling NAD in the cytosol, with most other mammalian cells using the malate-aspartate

shuttle for this purpose (*Mráček et al., 2013*; *Spinelli and Haigis, 2018*). Thus, we find that T-IEL, far from being metabolically quiescent, have instead a metabolism tailored to their environment, to protect T-IEL from the harsh intestinal environment and actively limit proliferation and activation of these cells.

Ribosomal content was the one area where T-IEL seemed truly deficient in comparison to naïve LN T cells. This was surprising, since naïve T cells, like T-IEL, are not actively cycling cells. However, it was recently shown that a subset of proteins in naïve T cells have short half-lives and are rapidly turned over (*Wolf et al., 2020*). These included transcription factors that maintain the naïve state, but that need to be rapidly degraded upon T cell activation, allowing T cells to differentiate. Naïve T cells were also found to express a large number of idling ribosomes ready to translate mRNAs required for T cell activation. Thus, the non-existent ribosomal activity in T-IEL subsets is possibly a reflection of their terminally differentiated status. It is also interesting to note that amino acid transporters were expressed at very low levels in T-IEL, thus limiting amino acid availability for protein translation. In this context, we recently showed that activation of T-IEL with IL-15 involves both upregulation of ribosome biogenesis and upregulation of amino acid transporters (*James et al., 2021*). The low rates of protein translation also highlight the importance of studying the proteome in T-IEL as there may be a significant disconnect between protein and mRNA expression.

In identifying proteins that were expressed solely in T-IEL, but not LN T cells, we uncovered a clear signature of T cell exhaustion within the proteome. Like exhausted T cells, T-IEL have diminished capacity to proliferate in response to TCR triggering and increased expression of co-inhibitory molecules. However, unlike exhausted T cells, T-IEL maintain high levels of cytotoxic effector molecules, suggesting that they are still capable of killing, although it is unclear what signals are required to trigger full degranulation in T-IEL. Interestingly, despite identifying more than 30 cell surface proteins on T-IEL that were not expressed in LN T cells, no one marker was exclusive to T-IEL, as they were either proteins that were normally expressed in intestinal epithelial cells or those expressed on activated or exhausted T cells. Thus, the proteomic profiles of T-IEL reveal an interesting mixture of various T cell types; naïve, effector, and exhausted, as well their unique tissue-specific signatures.

T-IEL also display several hallmarks of exhausted or suppressed T cells, including a major rewiring of the TCR signalosome. We were surprised to find that LAT2 and many negative regulators of signalling, such as DUSP6, were also expressed in induced T-IEL. These data suggest that the changes in the TCR signalosome are induced by the gut environment, rather than being developmentally regulated. In addition, we found that all T-IEL express ACAT2, a key protein involved in cholesterol esterification, that potentially sequesters cholesterol away from the plasma membrane. Previously, ACAT1, but not the closely related ACAT2, was found to be upregulated in activated CD8[+] T cells (*Yang et al., 2016*). Genetic ablation of ACAT1 lead to increased response from activated T cells in both infection and cancer, and this was attributed to the increased cholesterol in the plasma membrane leading to increased TCR clustering (*Molnár et al., 2012*; *Yang et al., 2016*). Indeed, increased cholesterol content has also been shown to potentiate γδ T cell activation (*Cheng et al., 2013*). It would be interesting to see if the high levels of ACAT2 expressed in T-IEL prevent accumulation of cholesterol in the T-IEL membranes, thus increasing the activation threshold of T-IEL. On a similar note, ARG2, which was highly expressed in T-IEL, has also been shown to block T cell activation (*Geiger et al., 2016*; *Martí i Líndez et al., 2019*), and expression of the alanine metabolising enzyme, GPT1, may limit alanine availability for protein synthesis in T cell activation (*Ron-Harel et al., 2019*). Thus, multiple lines of evidence support the notion that both natural and induced T-IEL are tightly regulated through inhibition of signalling.

In summary, we have presented an in-depth proteomic analyses and comparisons of induced and natural T-IEL and systemic T cells. These data provide key insights into the nature of T-IEL as well as the underappreciated similarities between both induced and natural T-IEL. New findings related to cholesterol metabolism, a high energy and translation barrier to activation, and transcription factors that potentially regulate T-IEL function suggest new ways to investigate how the different T-IEL subsets contribute to tissue and organismal homeostasis.

## Materials and methods

### Key resources table

| Reagent type (species) or resource | Designation | Source or reference | Identifiers | Additional information |
|---|---|---|---|---|
| Genetic reagent (*Mus musculus*) | P14 | PMID:2573841 | | |
| Strain, strain background (*Mus musculus*) | C57BL/6 J | Charles Rivers | RRID:IMSR_JAX:000664 | |
| Chemical compound, drug | DAPI | Thermo Fisher Scientific | Cat # D1306 | 1 µg/ml |
| Antibody | Anti-CD4 (rat, monoclonal) | Thermo Fisher Scientific (eBiosciences) | RRID:AB_494000 | Cell surface staining (1:200) |
| Antibody | Anti-CD8a (rat, monoclonal) | BioLegend | RRID:AB_2562558 | Cell surface staining (1:400) |
| Antibody | Anti-CD8a (rat, monoclonal) | BioLegend | RRID:AB_312746 | Immunofluorescence (1:100) |
| Antibody | Anti-CD8b (rat, monoclonal) | eBioscience | RRID:AB_657764 RRID:AB_1121888 | Cell surface staining (1:400) |
| Antibody | Anti-CD38 (rat, monoclonal) | BioLegend | RRID:AB_312928 RRID:AB_312929 | Cell surface staining (1:200) |
| Antibody | Anti-CD39 (rat, monoclonal) | BioLegend | RRID:AB_2563395 | Cell surface staining (1:200) |
| Antibody | Anti-CD44 (rat, monoclonal) | BD Biosciences | RRID:AB_1272244 | Cell surface staining (1:200) |
| Antibody | Anti-CD62L (rat, monoclonal) | Thermo Fisher Scientific | RRID:AB_469632 | Cell surface staining (1:200) |
| Antibody | Anti-CD73 (rat, monoclonal) | BioLegend | RRID:AB_11219608 | Cell surface staining (1:200) |
| Antibody | Anti-CD96 (rat, monoclonal) | BioLegend | RRID:AB_1279389 | Cell surface staining (1:200) |
| Antibody | Anti-CD103 (Armenian hamster, monoclonal) | BioLegend | RRID:AB_2563691 | Cell surface staining (1:200) |
| Antibody | Anti-CD160 (rat, monoclonal) | BioLegend | RRID:AB_10960740 RRID:AB_10960743 | Cell surface staining (1:200) |
| Antibody | Anti-CD244 (rat, monoclonal) | eBioscience | RRID:AB_657872 | Cell surface staining (1:200) |
| Antibody | Anti-E-cadherin (mouse monoclonal) | BD Biosciences | RRID:AB_397581 | Immunofluorescence (1:100) |
| Antibody | Anti-LAG-3 (rat, monoclonal) | eBioscience | RRID:AB_2573427 | Cell surface staining (1:100) |
| Antibody | Anti-P2 × 7 R (rat, monoclonal) | BioLegend | RRID:AB_2650951 | Cell surface staining (1:200) |
| Antibody | Anti-TCRb (Armenian hamster, monoclonal) | BioLegend | RRID:AB_2629696 | Cell surface staining (1:100) |
| Antibody | Anti-EpCam (rat, monoclonal) | eBioscience | RRID:AB_953617 | Cell surface staining (1:200) |
| Antibody | Anti-E-cadherin (rat, monoclonal) | eBioscience | RRID:AB_1834417 | Cell surface staining (1:100) |
| Antibody | Anti-TCRγδ (Armenian hamster, monoclonal) | BioLegend | RRID:AB_2563356 | Cell surface staining (1:200) |
| Antibody | Anti-phospho S6 (S235/236) (rabbit, monoclonal) | Cell Signaling Technology | RRID:AB_916156 | Intracellular staining (1:25) |
| Antibody | Anti-phospho ERK1/2 (T202/Y204) (rabbit, monoclonal) | Cell Signaling Technology | RRID:AB_331775 | Intracellular staining (1:200) |
| Antibody | Anti-ZO-2 (rabbit polyclonal) | Cell Signaling Technology | RRID:AB_2203575 | Immunofluorescence (1:50) |
| Antibody | Anti-CD3e (Armenian hamster, monoclonal) | BioLegend | RRID:AB_312667 | TCR stimulation (30 µg/ml) |
| Chemical compound, drug | PP2 | Merck (Calbiochem) | Cat # 529,573 | TCR stimulation, Src inhibitor |
| Chemical compound, drug | O-Propargyl-puromycin | JenaBioscience | NU-931–05 | Protein synthesis measurements |
| Commercial assay or kit | EasySep CD8$^+$ T cell isolation kit | STEMCELL Technologies, UK | Cat # 19,853 | For isolating CD8$^+$ T cells from LN |
| Commercial assay or kit | EasySep Mouse CD8a-positive selection kit II | STEMCELL Technologies, UK | Cat # 18,953 | For enriching CD8$\alpha^+$ IEL |

| Reagent type (species) or resource | Designation | Source or reference | Identifiers | Additional information |
|---|---|---|---|---|
| Commercial assay or kit | EasySep Dead Cell Removal (Annexin V) Kit | STEMCELL Technologies, UK | Cat # 17,899 | For removing dead epithelial cells and enriching IEL |
| Commercial assay or kit | Amplex Red cholesterol Assay Kit | Invitrogen | Cat # A12216 | Cholesterol assay |
| Commercial assay or kit | EZQ protein quantification kit | Thermo Fisher Scientific | Cat # R33200 | For accurate protein quantification for proteomics |
| Software, algorithm | MaxQuant | https://www.maxquant.org/ | RRID:SCR_014485 | Version 1.6.3.3 |
| Software, algorithm | Limma | *Ritchie et al., 2015* | RRID:SCR_010943 | Version 3.7 |
| Software, algorithm | qvalue | Bioconductor | RRID:SCR_001073 | Version 2.10 |
| Software, algorithm | FlowJo | TreeStar | | Version 10 |
| Software, algorithm | OMERO.figure | https://pypi.org/project/omero-figure/ | | Version 4.4.0 |
| Other | RPMI 1640 | Thermo Fisher Scientific/ GIBCO | 21875–034 | Media to culture cells |

## Mice

All mice were bred and maintained with approval by the University of Dundee ethical review committee in compliance with UK Home Office Animals (Scientific Procedures) Act 1986 guidelines. C57BL/6 J mice were purchased from Charles Rivers and acclimatised for a minimum of 10 days prior to use in experiments. Mice were maintained in a standard barrier facility on a 12 hr light/dark cycle at 21 °C, 45–65% relative humidity, in individually ventilated cages with corn cob and sizzler-nest material and fed an R&M3 diet (Special Diet Services, UK) and filtered water ad libitum. Cages were changed at least every 2 weeks. For all experiments, mice were used between 8 and 12 weeks of age, and for proteomics, male mice aged between 8 and 9 weeks were used.

## T-IEL and LN CD8 T cell isolation

T-IEL were isolated for sorting from mice and as described in *James et al., 2020*. Briefly, small intestines were extracted and flushed. Small intestines were longitudinally opened, then transversely cut into ~5 mm pieces and put into warm media containing 1 mM DTT. Small intestine pieces were shaken for 40 min, centrifuged, vortexed, and passed through a 100 μm sieve. The flow-through was centrifuged on a 36%/67 % Percoll density gradient at 700 g for 30 min. The T- I EL were isolated from the interface between 36% and 67% Percoll. In some experiments, isolated T-IEL were further enriched using an EasySep Mouse CD8α-positive selection kit (STEM-CELL Technologies) as per the manufacturer's instructions. Isolation and sorting details for the LN and effector populations used for proteomics can be found at http://www.immpres.co.uk/ under the 'Protocols & publications' tab.

## Proteomics sample preparation and peptide fractionation

Sample preparation was done as in *Howden et al., 2019*. Briefly, cell pellets were lysed, boiled, and sonicated, and proteins purified using the SP3 method (*Hughes et al., 2014*). Proteins were digested with LysC and trypsin and TMT labelling and peptide clean-up performed according to the SP3 protocol. The TMT labelling set-up is available in *Supplementary file 6*. The TMT samples were fractionated using off-line high-pH reversed-phase chromatography: samples were loaded onto a 4.6 mm × 250 mm Xbridge BEH130 C18 column with 3.5 μm particles (Waters). Using a Dionex BioRS system, the samples were separated using a 25 min multistep gradient of solvents A (10 mM formate at pH 9 in 2 % acetonitrile) and B (10 mM ammonium formate at pH 9 in 80 % acetonitrile), at a flow rate of 1 ml/min. Peptides were separated into 48 fractions, which were consolidated into 24 fractions. The fractions were subsequently dried, and the peptides were dissolved in 5 % formic acid and analysed by liquid chromatography–mass spectrometry.

## Liquid chromatography electrospray–tandem MS analysis

For each fraction, 1 µg was analysed using an Orbitrap Fusion Tribrid mass spectrometer (Thermo Fisher Scientific) equipped with a Dionex ultra-high-pressure liquid chromatography system (RSLC-nano). Reversed-phase liquid chromatography was performed using a Dionex RSLCnano high-performance liquid chromatography system (Thermo Fisher Scientific). Peptides were injected onto a 75 µm × 2 cm PepMap-C18 pre-column and resolved on a 75 µm × 50 cm RP C18 EASY-Spray temperature-controlled integrated column emitter (Thermo Fisher Scientific) using a 4 hr multistep gradient from 5 B to 35 % B with a constant flow of 200 nl/min. The mobile phases were: 2 % acetonitrile incorporating 0.1 % formic acid (solvent A) and 80 % acetonitrile incorporating 0.1 % formic acid (solvent B). The spray was initiated by applying 2.5 kV to the EASY-Spray emitter, and the data were acquired under the control of Xcalibur software in a data-dependent mode using the top speed and 4 s duration per cycle. The survey scan was acquired in the Orbitrap covering the *m/z* range from 400 to 1400 Thomson units (Th), with a mass resolution of 120,000 and an automatic gain control (AGC) target of $2.0 \times 10^5$ ions. The most intense ions were selected for fragmentation using collision-induced dissociation in the ion trap with 30 % collision-induced dissociation energy and an isolation window of 1.6 Th. The AGC target was set to $1.0 \times 10^4$, with a maximum injection time of 70 ms and a dynamic exclusion of 80 s. During the MS3 analysis for more accurate TMT quantifications, 10 fragment ions were co-isolated using SPS, a window of 2 Th and further fragmented using a higher-energy collisional dissociation energy of 55 %. The fragments were then analysed in the Orbitrap with a resolution of 60,000. The AGC target was set to $1.0 \times 10^5$ and the maximum injection time was set to 300 ms.

## MaxQuant processing

The raw proteomics data were analysed with MaxQuant (version 1.6.3.3) (*Cox and Mann, 2008*; *Tyanova et al., 2016*) and searched against a hybrid database. The database contained all murine SwissProt entries, along with TrEMBL entries with a human paralog annotated within human SwissProt and with protein-level evidence. The data was searched with the following modifications: carbamidomethylation of cysteine, as well as TMT modification on peptide amino termini and lysine side chains as fixed modifications; methionine oxidation and acetylation of amino termini of proteins were variable modifications. The FDR was set to 1 % at the protein and PSM level.

## Protein and BioReplicate filtering

Proteins groups marked as 'Contaminants', 'Reverse', or 'Only identified by site' were filtered out. Additionally, proteins detected with less than two unique and razor peptides were also filtered out.

Within both the TCRαβ CD8αα and TCRαβ CD8αβ T-IEL, one replicate (replicate 4) was filtered out from the downstream analysis due to protein content discrepancies. This biorep displayed a 15 % reduction in protein content compared to the other three replicates within the TCRγδ CD8αα and an increase in protein content of 32 % when compared to the remaining three replicates within the TCRαβ CD8αβ.

## Protein copy numbers and protein content

Protein copy numbers were estimated from the MS data using the proteomic ruler (*Wisniewski et al., 2014*) after allocating the summed MS1 intensities to the different experimental conditions according to their fractional MS3 reporter intensities. The protein content was calculated based on copy numbers. The molecular weight (in Da) of each protein was multiplied by the number of copies for the corresponding protein and then divided by $N_A$ (Avogadro's constant) to yield the individual protein mass in g/cell. The individual masses were converted into picograms and then summed for all proteins to calculate the protein content.

## Differential expression and overrepresentation analyses

All fold changes and p-values for the individual proteins were calculated in R utilising the bioconductor package LIMMA version 3.7. The Q-values provided were generated in R using the 'qvalue' package version 2.10.0. All other p-values were calculated using Welch's t-test. For all ORA, the background was set to the subset of proteins which were identified in either TCRαβ CD8αβ T-IEL or LN TCRαβ CD8αβ T cells. The GO ORAs were done using DAVID (*Jiao et al., 2012*) and PANTHER (*Mi et al., 2019*). Two distinct analyses were performed, one for proteins with a p-value < 0.001 and fold

change greater than or equal to the median+1.5 standard deviations and a second one for proteins with a p-value < 0.001 and fold change smaller than or equal to the median−1.5 standard deviations. The exhaustion ORA was done using WebGestalt (*Wang et al., 2017*) using the exhaustion markers provided reported within the literature (*Khan et al., 2019*) as a functional database for the analysis.

## Statistical significance thresholds

For the bar and box plots, symbols on bars represent independent biological replicates. For the MS derived bar plots and heatmaps; ** = p-value < 0.001 and fold change greater than or equal to the median+1 standard deviation, *** = p-value < 0.0001 and fold change greater than or equal to the median±1.5 standard deviations, based on the differential expression analyses described above. For all plots with non-MS based data, statistical analyses were carried out using R and GraphPad Prism version 8. The exact tests used are described in the figure legends, and p-values < 0.05 were considered significant.

## Flow cytometry

Cells were stained with titrated concentrations of the following murine monoclonal antibodies: TCRβ (clone H57–597 [BioLegend]), TCRγδ (clone GL3 [BioLegend or eBioscience]), CD4 (clone RM4-5 [BioLegend]), CD8α (clone 53–6.7 [BioLegend]), CD8β (clone H35–17.2 [eBioscience]), CD103 (clone 2E7 [BioLegend]), CD39 (clone Duha59 [BioLegend]), CD73 (clone Ty/11.8 [BioLegend]), CD38 (clone 90 [BioLegend]), P2 × 7 R (clone 1F11 [BioLegend]), CD244 (clone eBio244F4 [eBioscience]), LAG-3 (clone eBioC9B7W), CD160 (clone 7H1 [BioLegend]), CD96 (clone 3.3 [BioLegend]), EpCAM (clone G8.8 [eBioscience]), E-cadherin (clone DECMA-1 [eBioscience]), TOX (clone TXRX10 [eBioscience]). All data was acquired on an LSR Fortessa flow cytometer with DIVA software (BD Biosciences). Data were analysed using FlowJo software version 10 (TreeStar). For TCR stimulation, T-IEL and LN T cells were isolated and enriched for CD8$^+$ as described above. T-IEL were stained with Live/Dead fixable Near-IR (ThermoFisher) (1:250) for 10 min prior to stimulation, then combined with LN T cells at a 1:1 ratio and resuspended at a concentration of $10^6$ cells/ml in RPMI containing 1 % FBS, L-glutamine and penicillin/streptomycin. Cells were warmed at 37 °C before being stimulated with 30 µg/ml of anti-CD3 antibody (clone 145–2 C11 [BioLegend]) and 5 µg/ml of polyclonal anti-hamster cross-linking antibody (Jackson ImmunoResearch) for 5 min at 37 °C. For some samples as indicated, PP2 was added at a concentration of 20 µM for 1 hr prior to stimulation. After stimulation, cells were directly fixed in 2 % paraformaldehyde (PFA) 10 min at 37 °C before permeabilisation with 90 % ice-cold methanol. Samples were then fluorescently barcoded with different concentrations (0, 11.1, 33.3, or 100 µg/ml) of the Pacific Blue Dye (ThermoFisher) for 40 min, on ice before quenching with PBS + 0.5 % bovine serum albumin (BSA) (v:v). Barcoded samples were then pooled and stained for intracellular phospho-proteins, phospho-p44/42 MAPK (Erk1/2) (Thr202/Tyr204) (Clone 197G2 [Cell Signaling Technology]) and phospho-S6 ribosomal protein (Ser235/236) (Clone D57.2.2E [Cell Signaling Technology]) for 30 min at room temperature, protected from light, followed by secondary DyLight649 antibody (BioLegend). Cells were then stained for surface markers. Data were acquired using CytoFlex flow cytometer and analysed using FlowJo software (version 10). Data were analysed using the 'Forward deconvolution method' described in *Krutzik and Nolan, 2006*. Briefly, samples were differentiated based in the fluorescence intensities of each dye and then individual samples were analysed for their respective phospho-protein expression.

## Protein synthesis measurements

For comparing rates of protein synthesis, T-IEL and LN single-cell suspensions were cultured with 20 µM OPP (JenaBioscience) for 15 min. As a negative control, cells were pre-treated with 0.1 mg/ml CHX for 15 min before adding the OPP for 15 min (30 m total CHX exposure). Cells were then harvested, fixed with 4 % PFA, and permeabilised with 0.5 % Triton X-100 before undergoing a copper catalysed click chemistry reaction with Alexa 647-azide (Sigma). Following surface marker staining, cells were resuspended in PBS + 1 % BSA and analysed by flow cytometry to determine the degree of incorporation of OPP. All samples were acquired on an LSR Fortessa flow cytometer with DIVA software (BD Biosciences). Data were analysed using FlowJo software.

## Cellular cholesterol measurements

Cholesterol content was measured using the Amplex Red cholesterol Assay Kit (Invitrogen). T-IEL were sorted into TCRβ⁺ CD8αα⁺, TCRβ⁺ CD8αβ⁺, and TCRγδ⁺ CD8αα⁺ populations and LN cells were sorted for TCRβ⁺ CD8αβ⁺ CD44-lo CD62L-hi cells using the BD Influx Cell Sorter (BD Biosciences). Each population was lysed at a concentration of $10 \times 10^6$ cells/ml in 1× Amplex Red reaction buffer (Invitrogen) for 10 min at 4 °C then spun at 13,200 rpm for 12 min. Lysate was removed and diluted in 1× Amplex Red reaction buffer; 50 µl of 300 µM Amplex Red reagent (containing 2 U/ml horseradish peroxidase, 2 U/ml cholesterol oxidase, and 0.2 U/ml cholesterol esterase) was added to 50 µl of diluted lysate. The reaction was incubated for 30 min at 37 °C in the dark before reading on the Clariostar microplate reader (BMG Labtech) at excitation wavelength of 530 nm and emission wavelength of 590 nm. Cholesterol content in the lysates was calculated with reference to cholesterol standards.

## Immunofluorescence and imaging

T-IEL and LN T cells were isolated and enriched for CD8⁺ as described above. T-IEL were then washed twice in RPMI/10%FBS-containing media and further depleted of contaminating dead cells using EasySep Dead Cell Removal (Annexin V) Kit (STEMCELL Technologies) as per manufacturer's instructions, for negative enrichment of T-IEL. The purity and viability were checked by flow cytometry with CD8α-APC antibody and DAPI. LN CD8 T cells were at >98% purity, and the IEL were enriched to 60 % purity.

   $1 \times 10^6$ lymphocytes in PBS were gravity-sedimented onto each 18 mm round coverslip (N1.5) as described (*Tsang et al., 2017*). Cell were fixed in 3.7 % PFA in PBS, pH = 7.4, for 10 min at room temperature, washed with PBS, permeabilised with 0.5 % Triton X-100 in PBS for 15 min, washed with PBS, and blocked with 2 % BSA and 0.1 % Triton X-100 in PBS for 45 min prior to staining with primary antibodies diluted as indicated below in blocking solution, or blocking solution alone, for 1 hr 45 min at room temperature. Antibodies used were polyclonal rabbit anti-ZO-2 (Cell Signaling Technology) used at 1:50 dilution, mouse monoclonal anti-E-cadherin (BD Biosciences), used at 1:100 dilution, rat monoclonal anti-CD8α-FITC (clone 53–6.7 [Biolegend], used at 1:100). After three washes with PBS, cells were further incubated with appropriate secondary antibodies diluted 1:500 in blocking solution for 1 hr. Secondary antibodies used were goat anti-rat-AlexaFluor488 (Invitrogen); goat anti-mouse-AlexaFluor568 (Invitrogen); donkey anti-rabbit-AlexaFluor647 (Jackson ImmunoResearch). After additional three washes with PBS, cells were stained with 1 µg/ml DAPI in PBS for 10–15 min, washed with PBS, and mounted onto glass slides using ProLong gold antifade reagent (Thermo Fisher Scientific) as a mounting media. Stained cells were imaged using LSM 710 confocal microscope operated by Zen software (Zeiss) with a 63 ×/1.4NA oil immersion objective. For each image, 7–12 optical sections spanning the entire thickness of the cells were collected. The maximal intensity projections were generated and intensities adjusted in identical manner for all images in OMERO using the OMERO.figure app (*Allan et al., 2012*).

## Acknowledgements

The authors would like to thank Doreen Cantrell for her critical reading of the manuscript, support, and advice. We also acknowledge the support provided by A Whigham and R Clarke from the Flow Cytometry Facility for cell sorting and flow cytometry, and by the Biological Resources Unit at the University of Dundee. The contribution of the UK Research Partnership Infrastructure Fund award to the Centre for Translational and Interdisciplinary Research for the purchase of the mass spectrometers is gratefully acknowledged.

## Additional information

### Funding

| Funder | Grant reference number | Author |
|---|---|---|
| Wellcome Trust | 105024/Z/14/Z | Angus I Lamond |

| Funder | Grant reference number | Author |
|--------|------------------------|--------|
| Wellcome Trust | 206246/Z/17/Z | Mahima Swamy |
| Wellcome Trust | 215309/Z/19/Z | Olivia J James |
| Wellcome Trust | 222320/Z/21/Z | Harriet Watt |
| Wellcome Trust | 073980/Z/03/Z | Angus I Lamond |
| Wellcome Trust | 098503/E/12/Z | Angus I Lamond |

The funders had no role in study design, data collection and interpretation, or the decision to submit the work for publication.

## Author contributions

Alejandro J Brenes, Data curation, Formal analysis, Investigation, Methodology, Software, Writing – original draft; Maud Vandereyken, Data curation, Formal analysis, Investigation, Validation, Visualization, Writing – original draft; Olivia J James, Investigation, Validation, Writing – review and editing; Harriet Watt, Investigation, Validation; Jens Hukelmann, Formal analysis, Methodology; Laura Spinelli, Data curation, Investigation, Methodology, Writing – review and editing; Dina Dikovskaya, Validation; Angus I Lamond, Funding acquisition, Methodology, Project administration, Resources, Writing – review and editing; Mahima Swamy, Conceptualization, Data curation, Formal analysis, Investigation, Project administration, Supervision, Writing – original draft

## Author ORCIDs

Alejandro J Brenes http://orcid.org/0000-0001-8298-2463
Maud Vandereyken http://orcid.org/0000-0002-6169-7396
Olivia J James http://orcid.org/0000-0002-4294-7855
Harriet Watt http://orcid.org/0000-0002-4239-9051
Laura Spinelli http://orcid.org/0000-0002-5801-6297
Dina Dikovskaya http://orcid.org/0000-0003-3161-3072
Angus I Lamond http://orcid.org/0000-0001-6204-6045
Mahima Swamy http://orcid.org/0000-0003-3977-3425

## Ethics

All mice were bred and maintained with approval by the University of Dundee ethical review committee in compliance with U.K. Home Office Animals (Scientific Procedures) Act 1986 guidelines. C57BL/6J mice were purchased from Charles Rivers and acclimatised for a minimum of 10 days prior to use in experiments. Mice were maintained in a standard barrier facility on a 12hour light/dark cycle at 21°C, 45-65% relative humidity, in individually ventilated cages with corn cob and sizzler-nest material and fed an R&M3 diet (Special Diet Services, UK) and filtered water ad libitum. Cages were changed at least every two weeks. For all experiments, mice were used between 8-12 weeks of age, and for proteomics, male mice aged 8-9 weeks were used.

## Decision letter and Author response

Decision letter https://doi.org/10.7554/eLife.70055.sa1
Author response https://doi.org/10.7554/eLife.70055.sa2

# Additional files

## Supplementary files

• Supplementary file 1. Estimated protein copy numbers and differential expression analysis derived from the mass spectrometric proteomics data for the three tissue-resident intestinal intraepithelial T lymphocytes (T-IEL) subsets and wild-type (WT) and P14 lymph node (LN) T cells.

• Supplementary file 2. PANTHER gene ontology enrichment analysis.

• Supplementary file 3. DAVID functional annotation enrichment analysis.

• Supplementary file 4. Abbreviations and protein names. (related to *Figures 4–8*): Abbreviations and full protein names of proteins mentioned in the text and figures.

• Supplementary file 5. Exhaustion related protein lists. (related to *Figure 7e–f*): Proteins expressed in induced tissue-resident intestinal intraepithelial T lymphocytes (T-IEL) and found to be up-

regulated in exhausted T cells gene set, and proteins missing or downregulated in induced T-IEL, found to be down-regulated in exhausted T cells gene set (*Khan et al., 2019*).

- Supplementary file 6. Set-up of tandem mass tags (TMT) labelling of samples for proteomics.
- Transparent reporting form

### Data availability

The raw and processed mass spectrometry proteomics data have been deposited to the Proteome-eXchange Consortium via the PRIDE partner repository (Perez-Riverol et al., 2019) with the dataset identifier PXD023140 (https://www.ebi.ac.uk/pride/archive/projects/PXD023140/). All other data generated in this study are included within the manuscript and supporting files.

The following dataset was generated:

| Author(s) | Year | Dataset title | Dataset URL | Database and Identifier |
|---|---|---|---|---|
| Brenes AJ, Vandereyken M, James OJ, Spinelli L, Hukelmann J, Lamond AI, Swamy M | 2021 | Tissue adaptation is the dominant driver of the proteomic landscape of intestinal intraepithelial lymphocytes | https://www.ebi.ac.uk/pride/archive/projects/PXD023140/ | PRIDE, PXD023140 |

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
