## [Decision Letter]

[Editors' note: this paper was reviewed by Review Commons.]

**Acceptance summary:**

In this manuscript, Brenes and colleagues make use of quantitative mass spectrometry to compare the proteomes of induced tissue-resident intestinal intraepithelial T lymphocytes (T-IEL) and natural T-IEL subsets with naive CD8 T cells. Their findings reveal a dominant effect of the intestinal environment over the ontogeny on the T-IEL phenotypes. The results from this study should be a good resource to scientists interested in IELs specifically as well as those interested in the plasticity and adaptation of immune cells to the environments they reside in general.

---

## [Author Response]

We would like to thank the reviewers for their interest in our work and for their constructive comments and suggestions, which have strengthened our findings and improved our manuscript. A point-by-point response addressing each of their comments, and how our revised manuscript addresses the issues raised, is detailed below.

Reviewer #1 (Evidence, reproducibility and clarity (Required)):

Intestinal intraepithelial T lymphocytes (T-IEL) are a hetergenous T cell population with distinct developmental pathways. In the submitted manuscript, the authors applied high-resolution, quantitative proteomics to characterize subsets of T-IELs from mice. By comparing the proteomes of different T-IEL subsets as well as with naive CD8^+^ T cells from lymph nodes, they found more similarities between the T-IEL subsets than peripheral T cells, suggesting tissue microenvironment may play essential role to define intestinal T-IEL's identity. They further identified reduced protein synthesis, distinct metabolic profile and changes in T cell antigen receptor signalling pathways in T-IEL compared to peripheral T cells from LN. They also reported that T-IEL share some similarities with exhausted T cells. High quality proteomics data have been generated in this study, which can be served as resource for further studies for functional characterization of critical proteins to maintain intestine homeostasis or immunity. However, there are certain issues raised and need to be addressed before publication.

We thank the reviewer for recognising the quality of our data in this manuscript, and its value as a resource.

Major concerns:1. Lack of functional assays to characterize the proposed effects. All the claimed conclusions in this study are mostly based on proteomics data. Although OPP incorporation assay was performed to confirmed the reduced protein synthesis, but no statistical analysis was presented in the shown data (Figure 3F). Additionally, statistical analysis is missing in several figures, such as Figure 3, Figure 4B, 4D, Figure 6D, 6E. For Figure 7, whether all the presented selected proteins are statistically significantly changed? Therefore, it seems most of the conclusions are over stated. 

We apologise for the omission of statistical information. We have now added the requested p-values to all figures, including the OPP incorporation assay, which was indeed significantly different between T-IEL and naïve LN T cells.

Besides the OPP assay, validation of surface receptor expression of CD73, P2X7R, CD39, CD38, CD244, LAG3, CD160 and CD96 were also shown in Figure 6D, E (now Figure 7a-d), for cell surface receptor expression. To make it clearer that these figures were based on flow cytometry, we have included the flow plots of the different subsets.

We have now provided further functional validation of the proteome, including measurement of cholesterol levels in T-IEL compared to LN T cells (Figure 5c). These data show that all 3 T-IEL subsets contain up to 4-fold more cholesterol than LN T cells, supporting our finding that the cholesterol biosynthetic pathway is highly overexpressed in T-IEL. Further, immunofluorescence imaging of so-called epithelial proteins (ZO-2, E-Cadherin), and flow cytometry (EpCAM, E-Cadherin, TOX), show that T-IEL do indeed express epithelial proteins and the exhaustion-driving transcription factor TOX. These data are included in Figures 6, 7 and figure 6-supplement 1. Finally we have added new data to Figure 8, showing that both the natural T-IEL subsets are unresponsive to TCR stimulation, but the induced TCRab CD8ab population are highly responsive.

We have also included statistical information on all the heatmaps to make it clearer which proteins are differentially expressed in T-IEL by at least 1 standard deviation (~2.5 fold) significantly compared to LN T cells. With this information and the new validation, we hope the reviewer agrees that our data support our conclusions.

2. Purity of FACS sorted cells is only "about 50% to greater than 95%" (page 4). This shows big variation of cell purity, it's hard to accept 50% purity of FACS sorted cells. The purity should be improved. 

We apologise for this misunderstanding – what we meant was that before cell sorting the purity was 50%, and after cell sorting the purity was >95%. We have modified this sentence to read “CD8^+^ T-IEL subsets were purified from wild type (WT) murine small intestinal epithelial preparations to greater than 95% purity by cell sorting.”

3. What is the retionale to use naïve CD8^+^ T cells from P14 transgenic mice to compare with T-IEL subsets? 

We used P14 LN T cells mainly as a comparator for naïve LN T cells from WT mice- that is, as P14 are genetically and developmentally distinct, due to the expression of a transgenic TCR, they could have been completely different from polyclonal naïve CD8^+^ LN T cells. Yet WT naïve LN T cells share more similarity to P14 T cells than to WT IEL from the same genetic background. These findings are shown in Figure 1e and 1i. We have added new text to the first Results section to make this clearer.

4. It's a bit surprise to see less protein synthesis in T-IEL than naïve CD8^+^ T cells from LN. It'd be good to show the proportions of naïve and actived T cells in each subsets of T-IELs. This will help to understand. 

We agree with the reviewer that it is indeed surprising that T-IEL have lower protein synthesis rates than even naïve T cells. T-IEL bear many hallmarks of activated/memory T cells, including expression of CD44 and CD69 (Author response image 1). We speculate that the low metabolic activity/protein synthesis in these cells serves as an added barrier to spurious activation of these cells, however it would be interesting in the future to investigate how these levels are regulated.

**Author response image 1. sa2fig1:** Representative flow cytometric dot plots showing the expression of activation markers CD69 and CD44 in all 3 populations of T-IEL.

5. Text flow was not very well organized, e.g. cytoskeleton proteins have been described in 2 sections, in page 5, the 3rd paragraph (Figure 2) and page 8, the 4th paragraph (Figure 6). It'd be good to extract protein expression data from cytoskeleton proteins and present them together in one separate section. 

We thank the reviewer for pointing this out. Based on your suggestions we have streamlined the text to minimise repetition. We do still mention cytoskeletal proteins in Figure 2 as part of the global analysis and Gene ontology enrichment, however all detailed analyses of cytoskeletal proteins and names are collected into the section exploring cell surface receptors.

Minor concerns:1. Some texts in Results section should be moved to discussion, such as in page 6 about ARG2, instead of presenting data, there are too much speculations. 

We thank the reviewer for this recommendation, and have moved any speculative text to the discussion.

2. In page 7, it is claimed "T-IEL express the fatty acid transport proteins (FATP2(Slc27a2) and FATP4 (Slc27a4))". However, FATP2 is not presented in Figure 5C. It is also not clear whether FATP1, 3 and 4 are expressed significantly higher than in T-IEL? 

We thank the reviewer for catching this mistake. It was supposed to be FATP2 in the figure, not FATP3. We have changed this, and also added statistical information to the figure.

3. In page 6, "T-IEL express low levels (<2000 copies per cell) of 3 key amino acid transporters, i.e., SLC1A5, SLC7A5 and SLC38A2 (Figure 3C)". Based on Figure 3C, isn't SLC1A5 really expressed lower in T-IEL? Again, please show the statistical analysis results. 

We did not want to make a point that SLC1A5 was lower in T-IEL as compared to in LN T cells, only that the levels of these 3 amino acid transporters that are important for T cell activation are low. We have added the significance information to the graphs.

Reviewer #1 (Significance (Required)):SignificanceThe authors applied high-resolution, quantitative proteomics to characterize subsets of T-IELs from mice. The proteomics data presented are of high quality and can be served as a resource for further studies for functional characterization of critical proteins to maintain interest in homeostasis or immunity. However, this is more for basic immunological research, does not seem to have immediate translational potential. The authors also did not discuss such a potential. Given the data are generated from mice and conclusions made in this study are mostly just based on proteomics data, further functional characterization of the findings are largely missing. 

The reviewer questions the significance of our data since we explore murine T-IEL and not human T-IEL, therefore limiting translational potential. However, the intention of this manuscript is to trigger new research to explore the functional consequences of the differences between T-IEL and LN T cells, and to gain a deeper understanding into the regulation of these unique T lymphocytes. We hope that these future functional studies will have translational potential, as we did in our recent study (James OJ et al. (2021), Nature Communications, 12:4290) that identified PIM kinases are essential for IL-15 responses in murine T-IEL, and then found a similar upregulation of the PIM kinases in human Celiac disease patient samples, demonstrating the translational potential of murine T-IEL studies.

Reviewer #2 (Evidence, reproducibility and clarity (Required)):The research article titled "Tissue environment, not ontogeny, defines intestinal intraepithelial T lymphocytes" describes how various environmental signals shape the proteomes of three major tissue-resident intestinal intraepithelial T lymphocytes (T-IELs). The authors present a comprehensive proteomic approach to compare induced and natural T-IELs, as well as peripheral naïve CD8^+^ T cells derived from the lymph nodes (LNs).With regard to a localization of T-IELs (the interface between an antigen-rich lumen and gut tissue), it is not surprising that these cells are more defined by tissue environment than ontogeny. Although the observations raised by this manuscript are very interesting and some aspects are unexpected, there are several issues that should be addressed, without which clear conclusions cannot be made. 

We thank the reviewer for appreciating that our manuscript reveals novel and interesting aspects about T-IEL.

Major comments:There are some major concerns that the authors need to address before the manuscript can be further considered. Particularly, the expression of molecules in IELs that are normally produced by epithelial cells should be examined by other methods (e g. fluorescence microscopy etc.). The molecules such as Villin-1 and ZO-2 might originate from the contamination (intestinal epithelial cells?). Therefore, the authors have to explore these effects more thoroughly. 

We appreciate these concerns and have attempted to address them. We have now included immunofluorescence images showing that T-IEL do indeed express ZO-2 and E-Cad (Figure 6c), and flow cytometry data confirming the expression of E-Cadherin and EPCAM on T-IEL (Figure 6—figure supplement 1). Interestingly, analyses of microarray-based gene expression data from the ImmGen database (www.immgen.org), show that T-IEL express proportionately higher levels of mRNA for selected epithelial proteins, such as E-cadherin and ZO-2 (Figure 6—figure supplement 1). The fact that T-IEL have the mRNA for epithelial proteins supports our findings that these proteins are cell-intrinsic, and not contaminants identified by mass spectrometry.

We are gaining more insight into the T-IEL biology, but there is still a risk of over-interpretation of proteome-based data. Therefore, the core findings/most unexpected data should be further examined by functional approaches. E.g., the role of SREBP2 or LAT2 in T-IELs should be investigated by genetic deletion or knockdown of these molecules. Otherwise, these nice and unexpected findings would stay more descriptive. 

We thank the reviewer for these suggestions, and agree that further functional characterisation would be very insightful. However we feel that these are beyond the scope of this manuscript. Obtaining mice with genetic deletion of SREBP2 or LAT2 would take 6 months or longer, and then a further 6-8 months to obtain functional information. We believe rather that this is the strength and purpose of our manuscript, to encourage ideas such as these that open new avenues of research.

Minor comments: Although the figures are nicely presented, they look like figures from a review article. E.g., the Figure 5 and Figure 7 recapitulate our knowledge on lipid metabolism and T cell receptor signaling, respectively. They are more a schematic overview than figures originating from a research article. 

Here, we respectfully disagree with the reviewer, as we find it helpful to visualise the whole pathway and to see the protein expression data overlaid on the pathway. For example, in Figure 5a, depiction of the entire pathway makes it easier to see that every step of the cholesterol biosynthetic pathway is overexpressed in T-IEL. We have however modified the figures to make the proteomic data more prominent and have included p-values on them as well.

The authors should explicitly state that all conclusion made in the article are statements without functional data. 

We state at multiple points throughout the manuscript, including in the abstract, that our conclusions are based on analyses of proteomic data. We are also careful to only suggest functional consequences of our findings in the discussion, rather than assigning specific functions, and we end our manuscript with the statement that our data “suggest new ways to investigate how the different T-IEL subsets contribute to tissue and organismal homeostasis.” Therefore we feel we have sufficiently highlighted that our conclusions are based on proteomic data.

Reviewer #2 (Significance (Required)):There are several interesting and novel aspects in this study (a high significance in the research area of mucosal immunology). However, novel data should be performed by using functional approaches. Otherwise, there is a "danger" of over-interpretation of current results. The current version of the manuscript is rather a descriptive one. 

We believe the value of this manuscript is our in-depth analyses of the proteomes of T-IEL subsets, exploring how they differ from conventional T lymphocytes. These data and the ideas suggested by our analyses are to serve as a resource for the community, and to be thought-provoking for future research.

Reviewer #3 (Evidence, reproducibility and clarity (Required)):In this manuscript, Brenes et al., provide an exhaustive proteomic characterization of three different IEL populations: TCRabCD8aa, TCRabCD8ab, and TCRgdCD8aa cells. These cells are also compared to naïve CD8^+^ T cells derived from lymph nodes. These results indicate that these IEL populations, despite having different developmental pathways, they share many proteomic pathways. In addition, although these IEL populations are more similar to naïve peripheral T cells than to activated non-IEL, the different subpopulations of T-IEL resemble each other. The only major concern related to their results is the presence of proteins that are mostly associated with intestinal epithelial cells, such as villin. The authors indicate that the purity of their flow cytometric enrichment was between "50% to greater than 95%," which may account for the detection of proteins associated with epithelial cells. If that is the case, how is it possible for their overall data to be reliable? Can the authors elaborate on this point?Other than that concern, the manuscript is well written, follows a logical sequence and the data are interesting and compelling. 

We apologise for the misunderstanding– what we meant was that before cell sorting the purity was 50%, and after cell sorting the purity was >95%. We have modified this sentence to read “CD8^+^ T-IEL subsets were purified from wild type (WT) murine small intestinal epithelial preparations to greater than 95% purity by cell sorting.”

Given the high purity, we think that the epithelial proteins identified in T-IEL were not contaminants, but actually expressed in T-IEL. These data are now corroborated in the manuscript by inclusion of IF and flow cytometry data showing ZO-2, E-cadherin, EpCAM expression in T-IEL, and by mRNA expression data from ImmGen (see also response to reviewer 2) (Figure 6c and Figure 6—figure supplement 1). With these validations, our suggestion that T-IEL express multiple epithelial proteins as a means of communicating with epithelial cells is also better supported.

Reviewer #3 (Significance (Required)):This is the first report to my knowledge that describes in detail the proteomic profile of three different TCR+ IEL populations, which enhances the significance of this report. I believe, if the results are correct, this manuscript will be well received in the community of groups that study IEL biology. As an investigator that works with IEL, I believe this is an important manuscript with relevant data that will move the field forward. If my major concerns can be clarified, then I think this manuscript should be published.

We thank the reviewer for recognising the significance of our manuscript, and for their positive comments. We hope that they now find the revised manuscript suitable for publication.